# On Fitting Flow Models with Large Sinkhorn Couplings

## Abstract

Flow models transform data gradually from a modality (e.g. noise) onto another (e.g. images). Such models are parameterized by a time-dependent velocity field, trained to fit segments connecting pairs of source & target points. When a pairing between source and target points is known, the training boils down to a supervised regression problem. When no such pairing exists, as is the case when generating data from noise, training flows is much harder. A popular approach lies in picking in that case source and target points independently [Lipman et al., 2023]. This can, however, lead to velocity fields with high variance that are difficult to integrate. In theory, one would greatly benefit from training flow models by sampling pairs from an optimal transport (OT) measure coupling source and target, since this would lead to a highly efficient flow solving the Benamou and Brenier dynamical OT problem. Practically, recent works have proposed to sample *mini-batches* of $n$ source and $n$ target points and reorder them using an OT solver to form "better" pairs. These works have advocated using batches of size $n \approx 256$, and considered couplings that are both "hard" (permutations obtained with the Hungarian algorithm) or "soft" (computed with the Sinkhorn algorithm). We follow in the footsteps of these works by exploring the benefits of increasing this mini-batch size $n$ by several orders of magnitude, and look more carefully on the effect of the entropic regularization $\varepsilon$ used in Sinkhorn. Our analysis and computations are facilitated by new scale invariant quantities to present results and sharded computations parallelized over multiple GPU nodes. We uncover various dimensional regimes where flow matching benefits from OT guiding, using proper scales for $n$ and suitable entropic regularization $\varepsilon$, to be set so that it approximates $0.2$ in the novel renormalized entropy scale we propose.

## 1  Introduction

Finding a map that can transform a source into a target measure is a task at the core of generative modeling and unpaired modality translation. Following the widespread popularity of GAN formulations [Goodfellow et al., 2014], the field has greatly benefited from a gradual, time-dependent parameterization of these transformations as neural-ODEs [Chen et al., 2018] or normalizing flows [Rezende and Mohamed, 2015]. Such flow models are now commonly estimated using flow matching [Lipman et al., 2024]. While time parameterization substantially increases the expressivity of these models, this comes typically with a higher cost at inference time due to the additional cost of running an ODE solver with potentially dozens of steps. On the theoretical side, the golden standard for such time parameterized transformation is given by the Benamou and Brenier dynamical optimal transport (OT) solution, which would collapse in practice in a 1-step generation achieved by the Monge map formulation [Santambrogio, 2015]. In practice, while the mathematics [Villani, 2003] of optimal transport have contributed to the understanding of these methodsLiu et al., the jury seems to be still out on ruling whether tools from the computational OT toolbox [Peyré and Cuturi, 2019], which is typically used to compute large scale couplings from data [Klein et al., 2025], can decisively help with the estimation of flows in high-dimensions.

**Stochastic interpolants.** The flow matching (FM) framework [Lipman et al., 2024], introduced in concurrent seminal papers [Peluchetti, 2022, Lipman et al., 2023, Albergo and Vanden-Eijnden, 2023, Neklyudov et al., 2023] proposes to estimate a flow model by leveraging a time-dependent interpolation $\mu_t$ between source $\mu_0$ and target $\mu_1$ —the stochastic interpolant following the terminology of Albergo and Vanden-Eijnden [2023]. That interpolation is the crucial ingredient used to fit a parameterized velocity field, with a regression loss. In practice, such an interpolation can be formed by sampling $X_0 \sim \mu_0$ *independently* of $X_1 \sim \mu_1$, to define $\mu_t$ as the law of $X_t := (1-t)X_0 + tX_1$. One can then fit a parameterized time-dependent velocity field $\mathbf{v}_\theta(t, \mathbf{x})$ that minimizes the expectation of $\|X_1 - X_0 - \mathbf{v}_\theta(X_T, T)\|^2$ w.r.t. $X_0, X_1$ and $T$ a random time variable in $[0, 1]$. This procedure (hereafter abbreviated as Independent-FM, IFM) has been immensely successful, but can suffer from high variance, as highlighted by [Liu, 2022] (that loss can never be 0), and does not result in an optimal transport: this can be measured by noticing a high curvature when integrating the ODE needed to form an output from an input sample point $\mathbf{x}_0$.

**Blending FM and OT Solvers.** To fit exactly the OT framework, it would be best to choose $\mu_t$ to be the McCann interpolation between $\mu_0$ and $\mu_1$, which would be $\mu_t := ((1-t)\mathrm{Id} + tT^\star)\#\mu_0$, where $T^\star$ is the Monge map connecting $\mu_0$ and $\mu_1$. Unfortunately, this insight is irrelevant, since knowing $T^\star$ would mean that no flow needs to be trained at all. Adopting a more practical perspective, Pooladian et al. [2023] and Tong et al. [2023] have proposed in their seminal works to carefully select pairs of observations using OT solvers. Concretely, they sample mini-batches $\mathbf{x}_0^1 \ldots, \mathbf{x}_0^n$ from $\mu_0$ and $\mathbf{x}_1^1, \ldots, \mathbf{x}_1^n$ from $\mu_1$; compute a $n \times n$ OT coupling matrix; sample pairs of indices $(i_\ell, j_\ell)$ from that bistochastic matrix, and feed the flow model with pairs $\mathbf{x}_{i_\ell}^0, \mathbf{x}_{j_\ell}^1$. This approach was recently used and adapted in [Tian et al., 2024, Generale et al., 2024, Klein et al., 2023, Davtyan et al., 2025]. Despite their appeal, these modifications have not yet been widely adopted. The consensus stated recently by Lipman et al. [2024] seems to be still that *"the most popular class of affine probability paths is instantiated by the independent coupling"*.

**Can mini-batch OT really help?** We try to answer this question by noticing first that the evaluations carried out in all of the references cited above use batch sizes of $2^8 = 256$ points, more rarely $2^{10} = 1024$. We believe that this might be the case because these works rely on the Hungarian algorithm (complexity $O(n^3)$). We also notice that while these works also consider entropic OT (EOT) [Cuturi, 2013], they choose a single $\varepsilon$ value throughout their work. We go back to the drawing board in this paper, and study whether batch-OT FM can work at all, and if so at which regimes of mini-batch size $n$, regularization $\varepsilon$, and for which data dimensions $d$. Our contributions are:

- Rather than drawing a line between Batch-OT (in Hungarian or EOT form) and independent FM, we leverage the fact that *all* of these approaches can be interpolated using EOT: Hungarian corresponds to the case where $\varepsilon \to 0$ while IFM is recovered with $\varepsilon \to \infty$.

- We propose a modification of the Sinkhorn algorithm when used with for the squared-Euclidean norm, by dropping norms and focusing on the dot-product between points. We propose the definition of a renormalized entropy for couplings, to pin them efficiently on a scale of 0 (bijective assignment induced by a permutation, e.g. that returned by a Hungarian algorithm) to 1 (independent coupling). This quantity is useful because unlike transport cost or entropy regularization $\varepsilon$, it is bounded in $[0, 1]$ and is invariant to data dimension $d$ or number of points $n$.

- We explore in our experiments substantially different regimes for $n$ and $\varepsilon$. We vary the mini-batch size from $n = 2^{11} = 2,048$ to $n = 2^{21} = 2,097,152$ and consider a more ample adaptive grid for $\varepsilon$ that captures the range $[0, 1]$ range of our renormalized entropy.

## 2 Background Material on Optimal Transport and Flow Matching

Let $\mathcal{P}_2(\mathbb{R}^d)$ denote the space of probability measures over $\mathbb{R}^d$ with a finite second moment. Let $\mu, \nu \in \mathcal{P}_2(\mathbb{R}^d)$, and let $\Gamma(\mu, \nu)$ be the set of joint probability measures in $\mathcal{P}_2(\mathbb{R}^d \times \mathbb{R}^d)$ with left-marginal $\mu$ and right-marginal $\nu$. The OT problem in its Kantorovich formulation is:

$$W(\mu, \nu) := \inf_{\pi \in \Gamma(\mu, \nu)} \iint \tfrac{1}{2}\|x - y\|^2 \mathrm{d}\pi(x, y). \tag{1}$$

A minimizer of (1) is called an *OT coupling measure*, denoted $\pi^\star$. If $\mu$ was a source (e.g. noise) and $\nu$ a target (e.g. images), $\pi^\star$ would be the perfect coupling to sample pairs of noise and image to learn flow models: e.g. sample $\mathbf{x}_0, \mathbf{x}_1 \sim \pi^\star$ and ensure the flow models bring $\mathbf{x}_0$ to $\mathbf{x}_1$ along a straight path. Such of these couplings $\pi^\star$ are in fact induced by *pushforward maps*, i.e. a point $\mathbf{x}_0$ can only

be paired with a $T(\mathbf{x}_0)$, where $T : \mathbb{R}^d \to \mathbb{R}^d$. We say that such $T$ pushes $\mu$ forward to $\nu$, $T_{\#}\mu = \nu$, when for $X \sim \mu$ one has $T(X) \sim \nu$. The Monge formulation of OT is:

$$T^{\star}(\mu, \nu) := \underset{T : T_{\#}\mu = \nu}{\arg\min} \int \tfrac{1}{2}\|\mathbf{x} - T(\mathbf{x})\|^2 \mathrm{d}\mu(\mathbf{x}) \tag{2}$$

where the minimizers are referred to as Monge or OT maps. Such maps can be characterized:

**Theorem 1** ([Brenier, 1991]). *If $\mu \in \mathcal{P}_2(\mathbb{R}^d)$ has an absolutely continuous density then (2) is solved by a map $T^{\star}$ of the form $T^{\star} = \nabla u$, where $u : \mathbb{R}^d \to \mathbb{R}$ is convex. Moreover if $u$ is a convex potential that is such that $\nabla u_{\#}\mu = \nu$ then $\nabla u$ solves (2).*

As a result of this theorem, one can choose a convex potential $u$, a starting measure $\mu$, and train flow matching models between $\mu$ and $\nu := \nabla u_{\sharp}\mu$ to define synthetic tasks for which the coupling $\pi^{\star}$ is known, as proposed in [Korotin et al., 2021]. We consider this in Section 4.2 to benchmark batch-OT.

**Entropic OT.** Entropic regularization [Cuturi, 2013] has become the most popular approach to estimate a finite sample analog of $\pi^{\star}$ using samples $(\mathbf{x}_1, \ldots, \mathbf{x}_n)$ and $(\mathbf{y}_1, \ldots, \mathbf{y}_n)$. Using a regularization strength $\varepsilon > 0$, a cost matrix $\mathbf{C} := [\tfrac{1}{2}\|\mathbf{x}_i - \mathbf{y}_j\|^2]_{ij}$ between these samples, the entropic OT (EOT) problem can be presented in primal form (1) or in dual form:

$$\min_{\mathbf{P} \in \mathbb{R}_+^{n \times n}, \mathbf{P}\mathbf{1}_n = \mathbf{P}^T \mathbf{1}_n = \mathbf{1}_n/n} \langle \mathbf{P}, \mathbf{C}\rangle - \varepsilon H(\mathbf{P}), \quad \max_{\mathbf{f} \in \mathbb{R}^n, \mathbf{g} \in \mathbb{R}^n} \tfrac{1}{n}\langle \mathbf{f}+\mathbf{g}, \mathbf{1}_n\rangle - \varepsilon\langle \exp\left(\tfrac{\mathbf{C}-\mathbf{f}\oplus\mathbf{g}}{\varepsilon}\right), \mathbf{1}_{n \times n}\rangle. \tag{3}$$

The optimal solutions to (3) are usually found with the Sinkhorn algorithm, as presented in Algorithm 1, where for a matrix $\mathbf{S} = [\mathbf{S}_{i,j}]$ we write $\min_{\varepsilon}(\mathbf{S}) := [-\varepsilon \log\left(\mathbf{1}^{\top} e^{-\mathbf{S}_{i,\cdot}/\varepsilon}\right)]_i$, and $\oplus$ is the tensor sum of two vectors, i.e. $\mathbf{f} \oplus \mathbf{g} := [\mathbf{f}_i + \mathbf{g}_j]_{ij}$. The optimal dual variables (3) $(\mathbf{f}^{\varepsilon}, \mathbf{g}^{\varepsilon})$ can then be used to instantiate a valid coupling matrix $\mathbf{P}^{\varepsilon} = \exp\left((\mathbf{C} - \mathbf{f}^{\varepsilon} \oplus \mathbf{g}^{\varepsilon})/\varepsilon\right)$, which approximately solves the finite-sample counterpart of (1). An important remark is that as $\varepsilon \to 0$, the solution $\mathbf{P}^{\varepsilon}$ converges to the optimal transport matrix solving 1, while

---

**Algorithm 1** SINK($\mathbf{X} \in \mathbb{R}^{n \times d}, \mathbf{Y} \in \mathbb{R}^{m \times d}, \varepsilon, \tau$)

1: $\mathbf{f}, \mathbf{g} \leftarrow \mathbf{0}_n, \mathbf{0}_m$.
2: $\mathbf{C} \leftarrow [\tfrac{1}{2}\|\mathbf{x}_i - \mathbf{y}_j\|^2]_{ij}, i \leq n, j \leq m$
3: **while** $\| \exp\left(\tfrac{\mathbf{C}-\mathbf{f}\oplus\mathbf{g}}{\varepsilon}\right)\mathbf{1}_m - \tfrac{1}{n}\mathbf{1}_n \|_1 < \tau$ **do**
4:     $\mathbf{f} \leftarrow \varepsilon \log \tfrac{1}{n}\mathbf{1}_n - \min_{\varepsilon}(\mathbf{C} - \mathbf{f} \oplus \mathbf{g}) + \mathbf{f}$
5:     $\mathbf{g} \leftarrow \varepsilon \log \tfrac{1}{n}\mathbf{1}_n - \min_{\varepsilon}(\mathbf{C}^{\top} - \mathbf{g} \oplus \mathbf{f}) + \mathbf{g}$
6: **end while**
7: **return** $\mathbf{f}, \mathbf{g}, \mathbf{P} = \exp\left((\mathbf{C} - \mathbf{f} \oplus \mathbf{g})/\varepsilon\right)$

---

$\mathbf{P}^{\varepsilon} \to \tfrac{1}{n^2}\mathbf{1}_{n \times n}$ as $\varepsilon \to \infty$. These two limiting points coincide with the *optimal assignment* matrix (or optimal permutation as returned e.g. by the Hungarian algorithm [Kuhn, 1955]), and the uniform independent coupling used implicitly in I-FM.

**Independent and Batch-OT Flow Matching.** A stochastic interpolant $\mu_t$ with law $X_t := (1 - t)X_0 + tX_1$ is used in flow matching to solve a regression loss $\min_{\theta} \mathbb{E}_{T,X_0,X_1}\|X_1 - X_0 - \mathbf{v}_{\theta}(X_T, T)\|^2$ where the expectation is taken w.r.t. $X_0 \sim \mu_0, X_1 \sim \mu_1$ and $T$ a random variable in $[0, 1]$. In I-FM, this interpolant is implemented by taking independent batches of samples $\mathbf{x}_0^1 \ldots, \mathbf{x}_0^n$ from $\mu_0, \mathbf{x}_1^1, \ldots, \mathbf{x}_1^n$ from $\mu_1$, and $t_1, \ldots, t_n$ time values sampled in $[0, 1]$, to form the loss values $\|\mathbf{x}_1^k - \mathbf{x}_0^k - \mathbf{v}_{\theta}((1 - t_k)\mathbf{x}_0^j + t_k\mathbf{x}_1^k, t_k)\|^2$. In the

---

**Algorithm 2** FM 1-Step($\mu_0, \mu_1, n$, OT-SOLVE)

1: $\mathbf{X}_0 = (\mathbf{x}_0^1, \ldots, \mathbf{x}_0^n) \sim \mu_0$
2: $\mathbf{X}_1 = (\mathbf{x}_1^1, \ldots, \mathbf{x}_1^n) \sim \mu_1$
3: $\mathbf{P} \leftarrow$ OT-SOLVE($\mathbf{X}_0, \mathbf{X}_1$) or $\mathbf{I}_n/n$
4: $(i_1, j_1), \ldots, (i_n, j_n) \sim \mathbf{P}$
5: $t_1, \ldots, t_n \leftarrow$ TIMESAMPLER
6: $\tilde{\mathbf{x}}^k \leftarrow (1 - t_k)\mathbf{x}_0^{i_k} + t_k\mathbf{x}_1^{j_k}$, for $k \leq n$
7: $\mathcal{L}(\theta) = \sum_k \|\mathbf{x}_1^{j_k} - \mathbf{x}_0^{i_k} - \mathbf{v}_{\theta}(\tilde{\mathbf{x}}^k, t_k)\|^2$
8: $\theta \leftarrow$ GRADIENT-UPDATE($\nabla\mathcal{L}(\theta)$)

---

formalism of Pooladian et al. [2023] and Tong et al. [2023], the same samples $\mathbf{x}_0^1 \ldots, \mathbf{x}_0^n$ and $\mathbf{x}_1^1, \ldots, \mathbf{x}_1^n$ are first fed into a discrete optimal matching solver. This outputs a bistochastic coupling matrix $\mathbf{P} \in \mathbb{R}^{n \times n}$ which is then used to *re-shuffle* the $n$ pairs originally provided to be better coupled, and which should help the velocity field fit better trajectories, with less training steps. The procedure is summarized in Algorithm 2 and adapted to our setup and notations. The choice $\mathbf{I}_n/n$ corresponds to IFM. More recently, [Davtyan et al., 2025] has proposed to keep a memory of that matching effort across mini-batches, by updating a large (of the size of the entire dataset) assignment permutation between noise and full-batch data that is locally refreshed with the output of the Hungarian method run on a small batch. A crucial aspect of the batch-OT methodology is that this pairing is disconnected from the training of $\mathbf{v}_{\theta}$ itself. Indeed, as currently implemented, OT variants of FM can be interpreted as meta-dataloaders that do a selective pairing of noise and data, without considering $\theta$ at all. In that sense, training and preparation of coupled noise/data pairs can be done independently.

## 3 Prepping Sinkhorn for Large Batch-size and Dimension.

**The Necessity of Large Batch Size.** The motivation to use larger batch sizes for OT-FM lies in the fundamental bias introduced by using small batches in the context of the curse of dimensionality [Chewi et al., 2024, Fatras et al., 2019]. That bias cannot be traded off with more iterations on the flow matching loss. The necessity of varying $\varepsilon$ accordingly is that this regularization is known to offset that bias to some extent, with more favorable sample complexity [Genevay et al., 2018, Mena and Niles-Weed, 2019, Rigollet and Stromme, 2025].

**Automatic Rescaling of $\varepsilon$.** A practical problem arising when running the Sinkhorn algorithm lies in choosing the $\varepsilon$ parameter. As described earlier, while $\mathbf{P}^\varepsilon$ does follow a path from the optimal permutation return by the Hungarian algorithm to the independent coupling as $\varepsilon$ varies from $0 \infty$, what matters is what actual values are chosen in between those two ends. To avoid using a fixed grid that risks becoming irrelevant as we move $n$ and $d$, we revisit the strategy used in [Cuturi, 2013] to divide the cost matrix $\mathbf{C}$ by its mean, median or maximal value, as implemented for instance in [Flamary et al., 2021]. While needed to avoid underflow when instantiating a kernel matrix $\mathbf{K} = e^{-\mathbf{C}/\varepsilon}$, that strategy is not relevant when using the log-sum-exp operator in our implementation (as advocated in [Peyré and Cuturi, 2019, Remark 4.23]), since the $\min_\varepsilon$ in our implementation is *invariant* to a constant shift in $\mathbf{C}$, whereas mean, median and max statistics are not. We propose instead to use the *standard deviation* (STD) of the cost matrix, which has this property: dispersion of costs around its mean has more relevance than mean itself. The STD can be computed in $(n + m)d^2$ time/ memory, without having to instantiate the cost matrix. When this memory cost increase from $d$ to $d^2$ is too high, we subsample $n = 2^{14} = 16384$ points. In what follows, we always pass the $\varepsilon$ value to the Sinkhorn algorithm 1 as $\tilde{\varepsilon} := \mathrm{std}(\mathbf{C}) \times \varepsilon$, where $\varepsilon$ is now a scale-free quantity selected in a grid $[0.001, 1.0]$. See appendix for plots that report instead $\varepsilon$.

**Scale-Free Renormalized Coupling Entropy.** While useful to keep computations stable across runs, the rescaling of $\varepsilon$ still does not provide a clear idea of whether a computed coupling $\mathbf{P}^\varepsilon$ between $n \times n$ points is sharp or close to independent. While a distance to the independent coupling can be easily computed, that to the optimal Hungarian permutation cannot, of course, be derived. Instead, we resort to a fundamental information inequality used in [Cuturi, 2013]: if $\mathbf{P}$ is a valid coupling between two marginal probability vectors $\mathbf{a}, \mathbf{b}$, then one has by $\frac{1}{2}(H(\mathbf{a}) + H(\mathbf{b})) \leq H(\mathbf{P}) \leq H(\mathbf{a}) + H(\mathbf{b})$. As a result, for any $\varepsilon$, we can define a *renormalized* entropy $\mathcal{E}$ for any coupling of $\mathbf{a}, \mathbf{b}$:

$$\mathcal{E}(\mathbf{P}) := \frac{2H(\mathbf{P})}{H(\mathbf{a}) + H(\mathbf{b})} - 1 \in (0, 1].$$

When $\mathbf{a} = \mathbf{b} = \mathbf{1}_n/n$, as considered here, this simplifies to $\mathcal{E}(\mathbf{P}) := H(\mathbf{P})/\log n - 1$. Independently of the size $n$ and of the scale of $\varepsilon$, $\mathcal{E}(\mathbf{P}^\varepsilon)$ provides a simple measure of the proximity of $\mathbf{P}^\varepsilon$ to an optimal assignment matrix (as $\mathcal{E}$ gets closer to 0) or to the independent coupling matrix (as $\mathcal{E}$ reaches 1). As a result we report $\mathcal{E}(\mathbf{P}^\varepsilon)$ rather than $\varepsilon$ in our plots (or to be more accurate, the *average* of $\mathcal{E}(\mathbf{P}^\varepsilon)$ computed over multiple mini-batches).

**From Squared Euclidean Costs to Dot-products** Using the notation $T^\star(\mu, \nu)$ introduced in (2), we notice an equivariance property of Monge maps. For $\mathbf{s} \in \mathbb{R}^d$ and $r \in \mathbb{R}_+$ we write $L_{r,\mathbf{s}}$ for the dilation and translation map $L_{r,\mathbf{s}}(\mathbf{x}) = r\mathbf{x} + \mathbf{s}$. Naturally, $L_{r,\mathbf{s}}^{-1}(\mathbf{x}) = (\mathbf{x} - \mathbf{s})/r = L_{1/r, -\mathbf{s}/r}(\mathbf{x})$, but also $L_{r,s} = \nabla w_{r,s}$ where $w_{r,s}(\mathbf{x}) := \frac{r}{2}\|\mathbf{x}\|^2 - \mathbf{s}^T\mathbf{x}$ is convex.

**Lemma 2.** *The Monge map $T(\mu, \nu)$ is equivariant w.r.t to dilation and translation maps, as*

$$T^\star(L_{r,\mathbf{s}}\#\mu, L_{r',\mathbf{s}'}\#\nu) = L_{r',\mathbf{s}'} \circ T^\star(\mu, \nu) \circ L_{r,\mathbf{s}}^{-1}.$$

*Proof.* Following Brenier's theorem, let $u$ be a convex potential from $\mu$ to $\nu$ such that $T^\star(\mu, \nu) = \nabla u$. Set $F := L_{r',\mathbf{s}'} \circ \nabla u \circ L_{r,\mathbf{s}}^{-1}$. $F$ is the composition of the gradients of three convex functions. Because the Jacobians of $L_{r,\mathbf{s}}$ and $L_{r,\mathbf{s}}^{-1}$ are respectively $r\mathbf{I}_d$ and $\mathbf{I}_d/r$, they commute with the Hessian of $u$. Therefore the Jacobian of $F$ is symmetric positive definite, and $F$ is the gradient of a convex potential that pushes $L_{r,\mathbf{s}}\#\mu$ to $L_{r',\mathbf{s}'}\#\nu$. It is therefore their Monge map by Brenier's theorem. $\square$

In practice, this equivariance means that when focusing on permutation matrices (which can be seen as the discrete counterparts of these Monge maps), one is free to rescale and shift either point cloud. This remark has a practical implication when running Sinkhorn as well. When using the squared-Euclidean distance matrix, the cost matrix is a sum of a correlation term with two rank-1

norm terms, $\mathbf{C} = -\mathbf{X}\mathbf{Y}^T \frac{1}{2}(\xi\mathbf{1}_m^T + \mathbf{1}_n\gamma^T)$ where $\xi$ and $\gamma$ are the vectors composed of the $n$ squared norms of vectors in $\mathbf{X}$ and $\mathbf{Y}$. Yet, due to the constraints $\mathbf{P}\mathbf{1}_m = \mathbf{a}, \mathbf{P}^T\mathbf{1}_n = \mathbf{b}$, any modification to the cost matrix of the form $\tilde{\mathbf{C}} = \mathbf{C} - \mathbf{c}\mathbf{1}_m^T - \mathbf{1}_n\mathbf{d}^T$, where $\mathbf{c} \in \mathbb{R}^n, \mathbf{d} \in \mathbb{R}^m$ only shifts the (3) objective by a constant, $\langle \mathbf{P}, \tilde{\mathbf{C}} \rangle = \langle \mathbf{P}, \mathbf{C} \rangle - \frac{1}{n}\mathbf{1}_n^T\mathbf{c} - \frac{1}{n}\mathbf{1}_n^T\mathbf{d}$. In practice, this means that norms can only perturb Sinkhorn computations, and one should focus on the negative correlation matrix $\mathbf{C} := -\mathbf{X}^T\mathbf{Y}$, replacing Line 2 in Algorithm 1. We do observe significant stability gains of these properly rescaled costs when comparing two point clouds (see Appendix A.1).

**Scaling Up Sinkhorn to Millions of High-Dimensional Points.** Our ambition, when guiding flow matching with batch-OT as presented in Algorithm 2, is to vary $n$ and $\varepsilon$ so that the coupling $\mathbf{P}^\varepsilon$ used to sample indices can be both large ($n \approx 10^6$) and sharp if needed, i.e. with a $\varepsilon$ that can be brought to arbitrarily low levels so that $\mathcal{E}(\mathbf{P}^\varepsilon) \approx 0$. To that end, we leverage the OTT-JAX implementation of the Sinkhorn algorithm [Cuturi et al., 2022], which can be natively sharded across multi-GPUs, or more generally multiple nodes of GPU machines equipped with efficient interconnect. In that approach, inspited by the earlier mono-GPU implementation of [Feydy, 2020], all $n$ points from source and target are sharded across GPUs and nodes (we have used either 1 or 2 nodes of 8 GPUs each, either Nvidia H100 or A100). A crucial point in that implementation is that the cost matrix $\mathbf{C} = -\mathbf{X}\mathbf{Y}^T$ (following remark above) is never instantiated globally, and recomputed instead at each $\min_\varepsilon$ operation in Lines 4 and 5 of Algorithm 1 locally, for these shards. All sharded results are then gathered to recover $\mathbf{f}, \mathbf{g}$ newly assigned after that iteration. When outputted, we use $\mathbf{f}^\varepsilon$ and $\mathbf{g}^\varepsilon$ and, analogously, never instantiate the full $\mathbf{P}^\varepsilon$ matrix (this would be impossible at sizes $n \approx 10^6$ we consider) but instead, materialize it blocks of rows by blocks of rows to do index sampling. We use the Gumbel-softmax trick to vectorize and speed up efficiently the $n$ categorical sampling of these potentially very large unnormalized probability vectors.

# 4   Experiments

We revisit the application of Algorithm 2 using the modifications to the Sinkhorn algorithm outlined in Section 3 to various I-FM benchmark tasks. We consider synthetic tasks in which the ground-truth Monge map is known, and benchmark unconditioned image generation using CIFAR-10 and ImageNet-32 generation, with a limited number of total integration steps.

**Sinkhorn Hyperparameters.** To track precisely whether the Sinkhorn algorithm converges for low $\varepsilon$ values, we set the maximal number of iterations to $50,000$. We use the momentum rule introduced in [Lehmann et al., 2022] beyond 2000 iterations to speed-up harder runs. Overall, all of the runs below converge, and therefore, even for low $\varepsilon$, we never experience convergence issues. The threshold $\tau$ is set to 0.001 and we observe that it remains relevant for all dimensions, as we use the 1-norm to quantify convergence. Convergence statistics are reported in Appendix A.2.

## 4.1   Evaluation Metrics for $\mathbf{v}_\theta$

All metrics used in our experiments can be interpreted as *lower is better*. **Negative log-likelihood.** Given a trained flow model $\mathbf{v}_\theta(t, \mathbf{x})$, the density $p_t(\mathbf{x})$ obtained by pushing forward $p_0(\mathbf{x})$ along the flow map of $\mathbf{v}_\theta$ can be computed by solving

$$\log p_t(\mathbf{x}_t) = \log p_0(\mathbf{x}_0) - \int_0^1 (\nabla_x \cdot \mathbf{v}_\theta)(t, \mathbf{x}_t)\, \mathrm{d}t, \qquad \dot{\mathbf{x}}_t = \mathbf{v}_\theta(t, \mathbf{x}_t), \tag{4}$$

Similarly, given a pair $(t, \mathbf{x})$, the density $p_t(\mathbf{x})$ can be evaluated by backward integration [Grathwohl et al., 2018, Section 2.2]. The divergence $(\nabla_x \cdot \mathbf{v}_\theta)(t, \mathbf{x}_t)$ requires computing the trace of the Jacobian of $\mathbf{v}_\theta(t, \cdot)$. As commonly done in the literature, we use the Hutchinson trace estimator with a varying number of samples to speed up that computation without materializing the entire Jacobian and use either an Euler solver with 50 steps for synthetic tasks or a Dopri5 adaptive solver for image generation tasks, both implemented in the Diffrax toolbox [Kidger, 2021]. Given $n$ points $\mathbf{x}_1^1, \ldots, \mathbf{x}_1^n \sim \nu$ and integrated backwards, the negative log-likelihood (NLL) of that set is

$$\mathcal{L}(\theta) := -\frac{1}{n}\sum_{i=1}^n \log p_1(\mathbf{x}_1^i).$$

subject to (4) and $p_0$ the law of $\mu$. We alternatively report the bits per dimension (BPD) statistic, which is $\mathcal{L}$ divided by $d\log 2$.

**Curvature.** We use the *curvature* of the field $\mathbf{v}_\theta$ as defined by [Lee et al., 2023]: for $n$ integrated trajectories $(\mathbf{x}_t^1, \ldots, \mathbf{x}_t^n)$ starting from samples at $t = 0$ from $\mu$, the curvature is defined as

$$\kappa(\theta) := \tfrac{1}{n} \sum_{i=1}^{N} \int_0^1 \|\mathbf{v}_\theta(t, \mathbf{x}_t^{(i)}) - (\mathbf{x}_1^{(i)} - \mathbf{x}_0^{(i)})\|_2^2 \mathrm{d}t,$$

where the integration is done with an Euler solver with 50 steps for synthetic tasks and the Dopri5 solver evaluated on a grid of 8 steps for image generation tasks. The smaller the curvature, the more the ODE path looks like a straight line.

**Reconstruction loss.** For synthetic tasks in Sections 4.2, we have access to the ground-truth transport map $T_0$ that generated the target measure $\nu$. In both cases, that map is parameterized as the gradient of a convex Brenier potential, respectively a piecewise quadratic function and an input convex neural network, ICNN [Amos et al., 2017]. For a starting point $\mathbf{x}_0$, we can therefore compute a *reconstruction loss* (a variant of the $\mathcal{L}^2$-UVP in Korotin et al. [2021]) as the squared norm of the difference between the true map $T^\star(\mathbf{x}_0)$ and the flow map $T_\theta$ obtained by integrating $\mathbf{v}_\theta(t, \cdot)$ (using 50 steps with a Euler solver), defined using $n$ points sampled from $\mu$ as

$$\mathcal{R}(\theta) := \tfrac{1}{n} \sum_{i=1}^{n} \|T_\theta(\mathbf{x}_0^i) - T_0(\mathbf{x}_0^i)\|_2^2.$$

**FID.** We report the FID metric [Heusel et al., 2017] in image generation tasks. For CIFAR-10 we use the train dataset of 50k images, for ImageNet-32 we subset a random set of 50k images from the train set. For generation we consider four integration solvers, Euler with 4, 8 and 16 steps and a Dopri5 solver from the Diffrax library [Kidger, 2021].

### 4.2   Synthetic Benchmark Tasks, $d = 32 \sim 256$

We consider in this section synthetic benchmarks of medium dimensionality ($d = 64 \sim 256$). In this evaluation, we prioritize these tasks in controlled settings over other data sources at similar dimensions (e.g. PCA reduced single-cell data [Bunne et al., 2024]) because we want to compute a ground-truth reconstruction loss, and therefore elucidate the impact of OT batch size $n$ and $\varepsilon$ on this important practical aspect in practical applications.

**Piecewise Affine Brenier Map.** The source is a standard Gaussian and the target is obtained by mapping it through the gradient of a potential, itself a (convex) piecewise quadratic function obtained using the pointwise maximum of $k$ rank-deficient parabolas:

$$u(\mathbf{x}) := \max_{i \leq k} u_i(\mathbf{x}) := \tfrac{1}{2}\|\mathbf{x}\|^2 + \tfrac{1}{2}\|\mathbf{A}_i(\mathbf{x} - \mathbf{m}_i)\|^2 - \|\mathbf{A}_i \mathbf{m}_i\|^2, \tag{5}$$

where $\mathbf{A}_i \sim \text{Wishart}(\tfrac{d}{2}, I_d), \mathbf{m}_i \sim \mathcal{N}(0, 3I_d), c_i \sim \mathcal{N}(0, 1)$ and all means are centered around zero after sampling. In practice, this yields a transport map of the form $\nabla u(\mathbf{x}) = \mathbf{x} + A_{i^\star}(\mathbf{x} - \mathbf{m}_{i^\star})$ where $i^\star$ is the potential selected for that particular $\mathbf{x}$ (i.e. the argmax in (5)). The correction $-\|\mathbf{A}_i \mathbf{m}_i\|^2$ is designed to ensure that these potentials are sampled equally when moving away from 0. The number of potentials $k$ is equal to $d/16$. Examples of this map are shown in Appendix A.3. We consider this setting in dimensions $d = 32, 64, 128, 256$.

**Korotin et al. Benchmark.** We use the set of pre-trained ICNNs introduced in [Korotin et al., 2021] along with their predefined Gaussian mixtures as sources. We consider the benchmark in $d = 32, 64, 128, 256$ using their checkpoints to generate the ground-truth maps. This problem setting is more challenging, however, since both the source *and* target distributions have multiple modes.

**Velocity Field Parameterization and Training.** The velocity fields are parameterized as MLPs with 5 hidden layers, of sizes 512 for $d = 32, 64$ and 1024 for $d = 128, 256$. Time in $[0, 1]$ is encoded using $d/8$ Fourier encodings. All models are trained with unpaired batches: the sampling in Line 1 of Algorithm 2 is done as $\mathbf{X}_0 \sim \mu$ while for Line 2, $\mathbf{X}_1 := T_0(\mathbf{X}_0')$ where $\mathbf{X}_0'$ is a new sample from $\mu$. All models are trained for 8192 steps, with effective batch sizes of 2048 samples to average a gradient, a learning rate of $10^{-3}$ (we tested with $10^{-2}$ or $10^{-4}$, both were either unstable or less efficient on a subset of runs). The model marked as ▲ in the plots is a flow model trained with *perfect* supervision, i.e. given *ground-truth paired samples* $\mathbf{X}_0 \sim \mu$ and $\mathbf{X}_1 := T_0(\mathbf{X}_0)$, provided in the correct order. I-FM is marked as ▼. For all other runs, we vary $\varepsilon$ (reporting renormalized entropy $\mathcal{E}$)

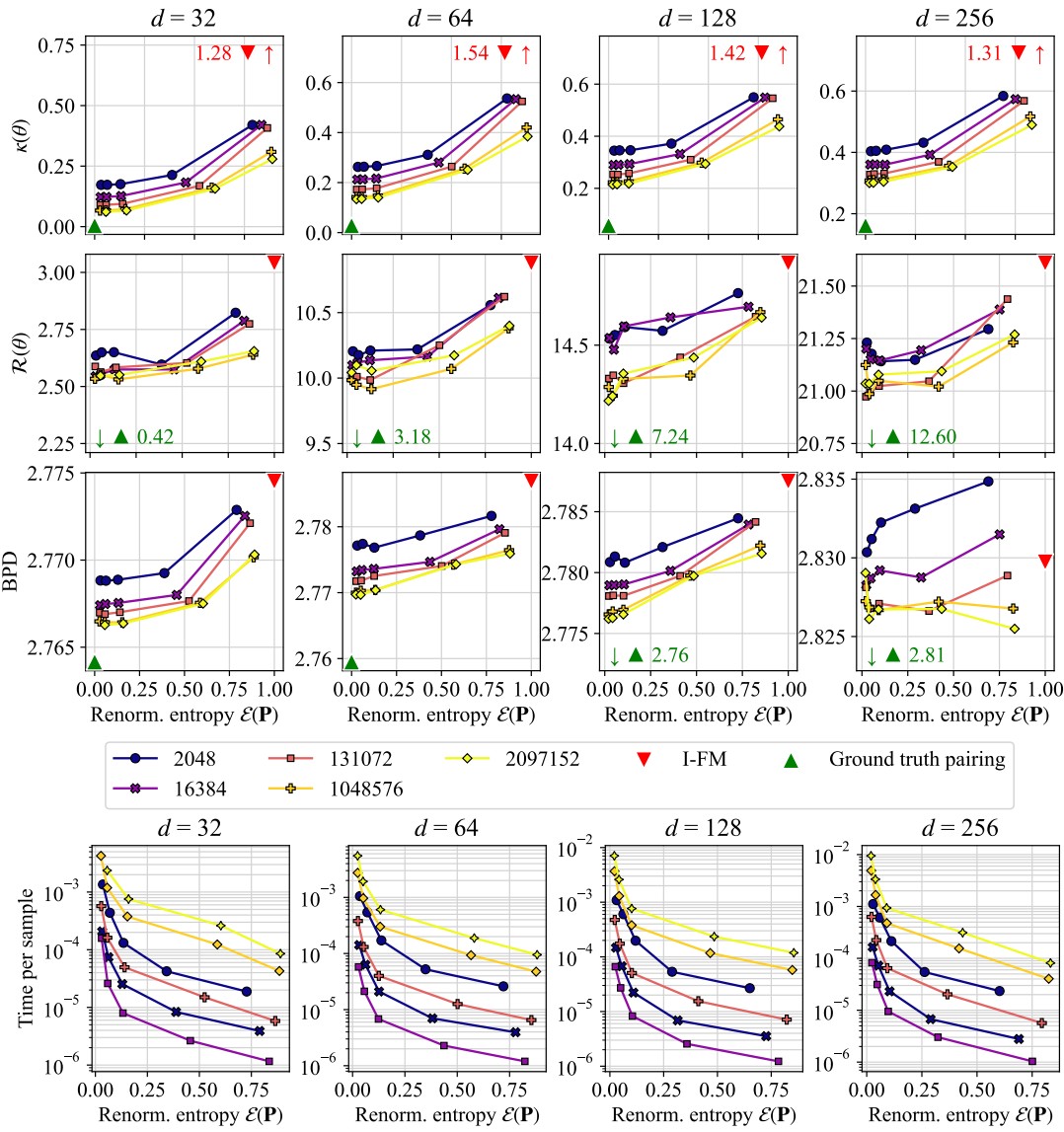

Figure 1: Results on the **piecewise affine OT Map benchmark**. The three top rows present (in that order) curvature, reconstruction and BPD metrics. Below, we provide compute times associated with running the Sinkhorn algorithm as a per-example cost. This per-example cost is the total time needed to run Sinkhorn to get $n \times n$ coupling divided by $n$. That cost would be 0 when using I-FM. We observe across all dimensions improvements of all metrics.

and the batch size $n$ used to compute couplings, somewhere between $256$ and $2,097,152$. These runs are carried out on a single node with 8 GPUs, and therefore the data is sharded in blocks of size $n/8$.

**Results.** The results displayed in Figures 1 and 2 paint a homogeneous picture: as can be expected, increasing $n$ is generally impactful and beneficial for all metrics. The interest of decreasing $\varepsilon$, while beneficial in smaller dimensions, can be less pronounced in higher dimensions. Indeed, we find that renormalized entropies around $\approx 0.2$ should be advocated, if one has in mind the computational effort needed to get these samples.

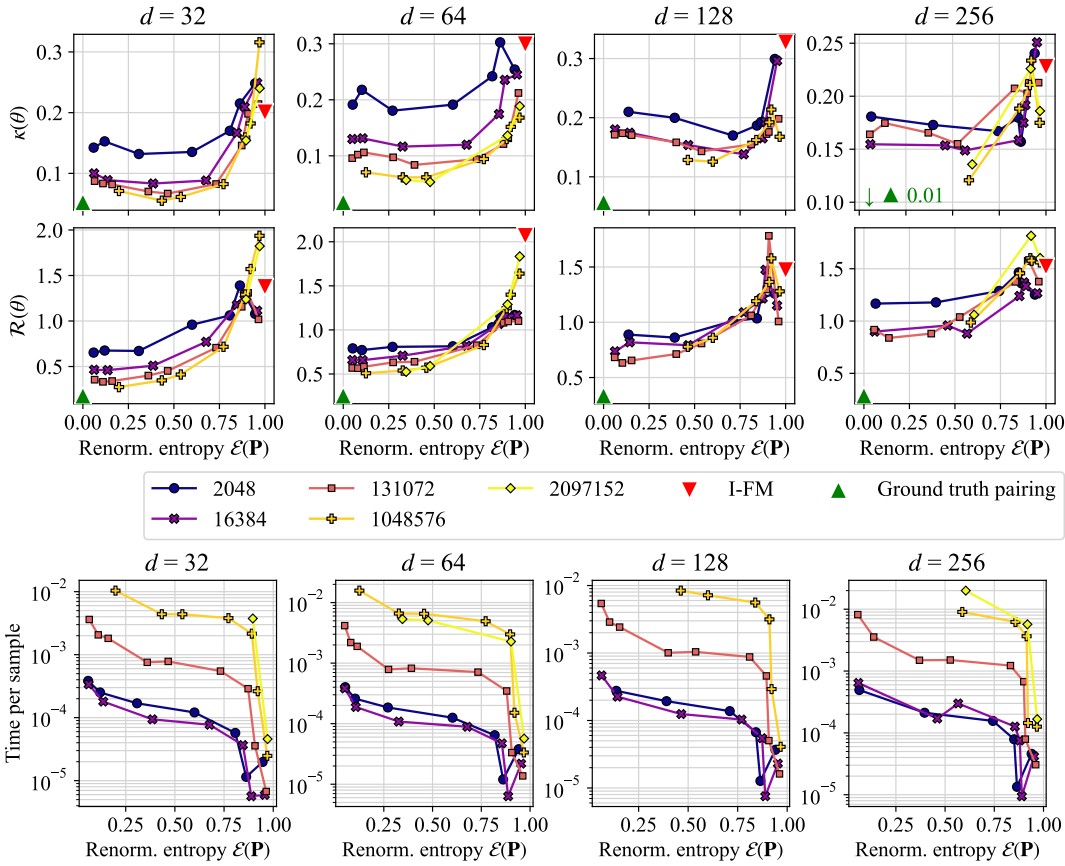

Figure 2: Results on the **Korotin benchmark**. As with Figure 1, we compute curvature and reconstruction metrics, and compute times below. Some of the runs for largest OT batch sizes $n$ are provided in the supplementary. These runs suggest that to train OT models in these dimensions increasing $n$ is overall beneficial across the board.

### 4.3 Unconditioned Image Generation, $d = 3072$

**CIFAR-10.** As done originally in [Lipman et al., 2023], we consider unconditional generation of the CIFAR-10 dataset. Results are presented in Figure 3. Compared to results reported in [Tong et al., 2023] we observe slightly better FID scores (about 0.1) for both I-FM and OT-FM.

**ImageNet-32.** As also considered in [Lipman et al., 2023], we also evaluate the impact of Batch-OT in unconditional generation of the ImageNet-32 dataset. We report results with under-trained models (120k steps vs. 438k advocated in their paper) in Figure 4 and present later checkpoints in Appendix A.6. Compared to results reported in [Tong et al., 2023] we observe slightly better FID scores (about 0.1 when using the Dopri5 solver for instance) for both I-FM and OT-FM.

**Velocity Field Parameterization and Training.** We use the network parameterization given in [Tong et al.] for CIFAR-10 and we replicate the network parameterization given in [Pooladian et al., 2023], including learning rate choices. We follow their recommendations on setting learning rates as well as total number of iterations.

**Limitations.** Our results rely on training of neural networks. In the interest of comparison, we have used the same model across all changes advocated in the paper (on $n$ and $\varepsilon$). However, and due to the scale of our experiments, we have not been able to ablate important parameters such as learning rates when varying $n$ and $\varepsilon$.

**Conclusion.** Our experiments suggest that guiding flow models with large scale Sinkhorn couplings can prove beneficial for downstream performance. We have tested this hypothesis by computing and sampling from both crisp and blurry $n \times n$ Sinkhorn coupling matrices for sizes $n$ in the millions of

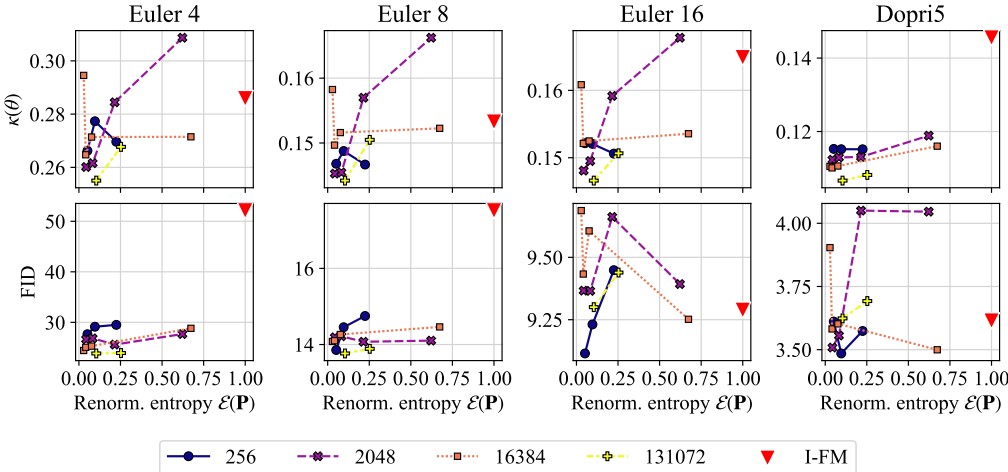

Figure 3: Experiment metrics for **CIFAR-10** image generation. We evaluate the trained models using the Euler solver with three different number of steps, and with the Dopri5 solver and adaptive steps. The plots demonstrate the benefits of a larger OT batch size to achieve significantly smaller curvature, and moderately smaller FID at low number of integration steps. Our experiments also suggest that in this setting, lower renormalized entropy generally benefits the performance.

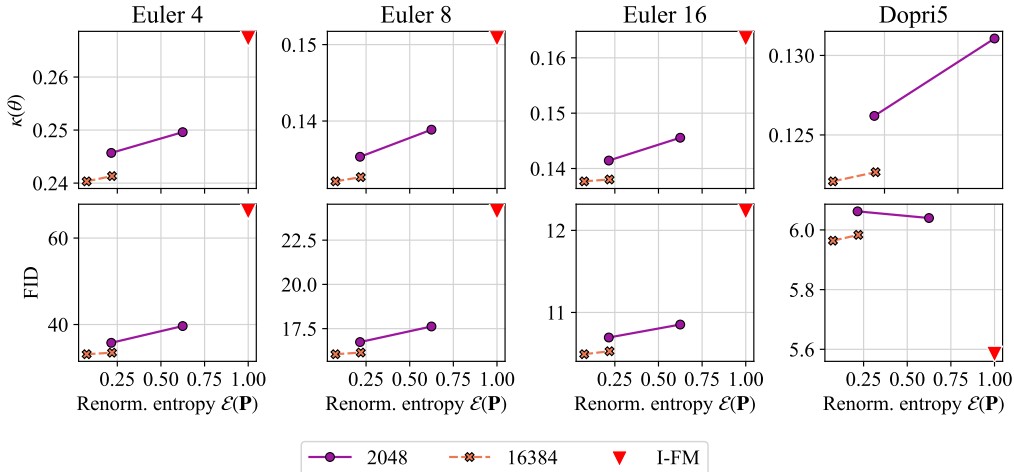

Figure 4: Early **ImageNet-32** experiment metrics obtained at a checkpoint of 120k iterations (150k for I-FM). We provide later checkpoint results and settings in Appendix A.6.

points, placing them on an intuitive scale from 0 (close to using an optimal permutation as returned e.g. by the Hungarian algorithm) to 1 (equivalent to the independent sampling approach popularized by Lipman et al. [2023]). This involved efficient multi-GPU parallelization, realizing scales which, to our knowledge, were never achieved previously in the literature. Although the scale of these computations may seem large, they are still relatively cheap compared to the price one has to pay to optimize the FM loss, and, additionally, are completely independent from model training. As a result, they should be carried out prior to any training. While we have not explored the possibility of launching multiple jobs with them (to ablate, e.g., for other fundamental aspects of model training such as learning rates), we leave a more careful tuning of these training runs for future work. We claim that paying this relatively small price to log and sample paired indices obtained from large scale couplings results for mid-sized problems in great returns in the form of faster training and faster inference, thanks to the straightness of the flows learned with the batch-OT procedure. For larger sized problems, the conclusion is not so clear, although we quickly observe benefits when using middle values for $n$ (in the thousands) and renormalized entropies around $0.2$ which forms, at the time of writing, our main practical recommendation for end users.

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

414 Please look at supplementary material zip file for appendix.


# A   Appendix / supplemental material

## A.1   Sinkhorn

Here put minimal evidence that dropping norm terms helps with convergence and considering smaller $\varepsilon$

## A.2   Sinkhorn Convergence

## A.3   Gaussian Generation

## A.4   Korotin et al. Benchmark Examples

## A.5   Cifar 10 Detailed Results

## A.6   ImageNet32 Detailed Results

