# OpenReview forum: "On Fitting Flow Models with Large Sinkhorn Couplings"
_NeurIPS.cc/2025/Conference — Submitted to NeurIPS 2025_

### Official Review · Reviewer_3ETv · 2025-06-15

**Clarity:** 3
**Significance:** 3
**Originality:** 3
**Rating:** 4
**Confidence:** 3

**Summary:**

This paper re-examines flow-based generative modeling by flow matching and investigates the benefit of using large-scale entropic optimal transport (EOT) couplings, i.e., Sinkhorn couplings, in order to match source and target samples. The authors think that the current practice uses small mini-batch sizes (e.g., 256) and fixed entropic regularization parameters $\epsilon$ which impose a ceiling on performance. The paper suggests:

* Using much bigger batch sizes (up to 2 million).

* Adding a batch-size and dimension-invariant, renormalized entropy scale, and facilitating more efficient tuning of $\epsilon$.

* Scaling the Sinkhorn algorithm efficiently across multi-GPU platforms using OTT-JAX.

* Demonstrating that couplings with low-entropy, large-batch values produce better training signals for flow models, minimizing curvature, reconstruction error, NLL, BPD, and FID scores across synthetic and real datasets (e.g., CIFAR-10, ImageNet-32).

* The method is meta-dataloader: training and generation coupling are separated, thus making the procedure modular and computationally realistic.

**Questions:**

* The paper [2]  proposes the two-stage approaches which utilizes RAM memory to scale up the OT problem in term of memory storage. Can this approach help to scale up further the OT problem in the paper?

[2] Improving Mini-batch Optimal Transport via Partial Transportation, Nguyen et al.

**Ethical Concerns:**

["NO or VERY MINOR ethics concerns only"]

**Final Justification:**

The authors addressed my questions.

**Limitations:**

Yes

**Quality:**

3

**Strengths And Weaknesses:**

# Strengths

* The paper has a clean motivation and an insightful problem formulation. The authors indicate the common drawback of previous work: small mini-batches and constant $\epsilon$ for EOT.

* The paper gives an original application of scale-free entropy
Definition of a renormalized scale of entropy,  provides an interpretable, dimension-invariant way to calibrate Sinkhorn couplings, dimension-invariant Sinkhorn coupling

* The paper provides massive-scale experiments i.e., 2M×2M coupling matrices are employed—a scale that has not yet appeared in the literature.

* The proposed methods give substantial practical gains i.e., experiments demonstrate consistent improvements over independent flow matching (IFM) on synthetic tasks and image generation, particularly on curvature and FID scores (Figures 1–4).

* The paper also provides an efficient implementation. The authors provide technical contributions in rescaling $\epsilon$ through std(C), replacing dot-products for Euclidean norms for stability, and sharding Sinkhorn computations.

# Weaknesses

* The role of coupling size and entropy in training stability or convergence speed is not adequately explored. Authors state they did not ablate learning rates for scale (p. 8, l. 280).

* While the paper claims the Sinkhorn step is relatively low-cost, 2M×2M couplings take 8+ GPUs and substantial memory (p. 5, l. 192–207), which will hold back adoption.

* Experiments lack missing standard deviation/error bars, which would help to estimate variability across runs (Checklist point 7, p. 13).

* The paper has not investigated the usage of other variants of OT which are shown to be better in a mini-batch setting [1] [2].

[1] Unbalanced minibatch Optimal Transport; applications to Domain Adaptation, Fatras et al.

[2] Improving Mini-batch Optimal Transport via Partial Transportation, Nguyen et al.

---

> ### Author Rebuttal · Authors · 2025-07-30
>
> >**The paper has a clean motivation and an insightful problem formulation **[…] The paper provides massive-scale experiments i.e., 2M×2M coupling matrices are employed—a scale that has not yet appeared in the literature […]**The proposed methods give substantial practical gains**
>
> Many thanks for your very encouraging comments!
>
> >**The role of coupling size and entropy in training stability or convergence speed is not adequately explored. Authors state they did not ablate learning rates for scale (p. 8, l. 280).**
>
> We are happy to report that learning rates have been ablated for Imagenet-32 and Imagenet-64. It seems that the original choices reported in the literature for IFM (`lr=1e-4`) are suitable for OTFM at all $n$ and $𝜀$ scales.
>
> While sharing plots would be more convenient, we cannot. Hence, here are two table describing FID along iterations for ImageNet-32. We do observe, for ImageNet-32, a decrease then increase of the FID metric along iterations, even for IFM and other settings --- this was mentioned in Table 1 of the supplementary. We do not observe this in our runs on ImageNet-64.
>
> ### $n = 1048576, 𝜀= 0.03$, FID@DOPRI5
> | Step | LR = 1e-4 (default) | 3e-4 (bigger) | 7e-5 (smaller) |
> | ------:| ------:| ------:| ------:|
> | 30001| 67.29 | 112.34 | 61.89|
> | 90003| 7.02 | 390.12 | 7.61 |
> | 150005 | 5.43 | 34.32| 5.89 |
> | 210007 | 5.00 | 30.47| 5.35 |
> | 270009 | **4.94** | 11.95| 5.24 |
> | 330011 | 5.16 | 9.77 | 5.47 |
> | 438015 | 5.88 | 9.25 | 6.19 |
>
> ### $n = 1048576, 𝜀= 0.03$, FID@4
> | Step | LR = 1e-4 (default) | 3e-4 (bigger) | 7e-5 (smaller) |
> | ------:| ----:| ---:| ----:|
> | 30001| 66.92 | 144.98 | 50.47|
> | 90003| 30.52 | 394.15 | 31.68|
> | 150005 | **29.52** | 58.67| 30.45|
> | 210007 | 29.65 | 51.21| 30.39|
> | 270009 | 30.01 | 34.60| 30.53|
> | 330011 | 30.57 | 35.07| 31.06|
> | 438015 | 31.43 | 35.26| 31.69|
>
>
>
> >**While the paper claims the Sinkhorn step is relatively low-cost, 2M×2M couplings take 8+ GPUs and substantial memory (p. 5, l. 192–207), which will hold back adoption.**
>
>
> This is a valid point, although 8-GPU nodes are by now the norm. We observe that most config files shared nowadays to train even small FM models on ImageNet-32 assume access to a node.
>
> In our response to Reviewer **zUcn** we have considered several ways to improve the picture in sections
> - `1. Warmstart` (to reduce overall the number of iterations),
> - `2. Computing matchings in PCA Space` (to alleviate per-iteration cost and lower memory requirements for couplings)
> - `3. Precomputing Noise/Data Pairs` to split preprocessing from training.
>
>
>
> >**Experiments lack missing standard deviation/error bars, which would help to estimate variability across runs (Checklist point 7, p. 13).**
>
>
> As mentioned in our response to W4 of Reviewer **7qcL**, we cannot realistically provide so many $n,𝜀$ results along with error bars, as this would make our compute budget explode.
>
> We notice that other researchers have recently complained specifically about this requirement (Ben Recht, _Standard error of what now?_, blog post) because it is not realistic.
>
>
> >**The paper has not investigated the usage of other variants of OT which are shown to be better in a mini-batch setting [1] [2]**.
>
>
> Thanks for these references! We have added these to the draft. However, we observe that:
>
> * These approaches explore partial or unbalanced OT, which are (for now) unrelated to our setting.
> * The code provided for these approaches cannot scale as they both rely on POT and GeomLoss solvers, which are intrinsically either CPU or single-GPU.
> * We also note that [2] claims, regarding Sinkhorn, in its introduction, “that storing an n × n matrix is unavoidable in this [Sinkhorn] approach, thus the memory capacity needs to match the matrix size”. Interestingly, we believe that this is incorrect, and this is one of the pillars of our contribution, as explained in L.200, see also our point `2. Computing matchings in PCA Space` to Reviewer **zUcn** for more details on online recomputations of cost matrices, a technique that was leveraged first by `GeomLoss`.
>
> As explained in our answer to point **S1**, Reviewer **52g8**, we do not see, at the moment, an alternative solver to the Sinkhorn algorithm that can simultaneously
> * be implemented to utilise efficiently / maximally all 8 GPU nodes, to scale to millions of points;
> * return couplings that can be arbitrarily close to the solution of exact OT;
>
> Of course, we hope that other researchers will be encouraged, after seeing our work, to target the 1 million point scale with other solvers / variants, and demonstrate their relevance to flow matching.

---

> > ### Comment · Reviewer_3ETv · 2025-08-02
> >
> > I would like to thank the authors for the response. I will keep my positive score.

---

### Official Review · Reviewer_Lgwo · 2025-06-16

**Clarity:** 3
**Significance:** 3
**Originality:** 2
**Rating:** 5
**Confidence:** 3

**Summary:**

The paper experimentally tests the potential advantages in initializing flow matching algorithms with Sinkhorn couplings instead of the independent coupling. That is, for a given regularization parameter $\varepsilon>0$, they sample $n$ source and $n$ target points $x^1_0,\cdots,x^n_0$ and $x^1_1,\cdots,x^n_1$, solve the corresponding entropic optical transport problem by means of Sinkhorn's iterations. The optimal coupling is then used a starting point for Flow matching instead of the independent coupling. The motivation for doing this lies in the fact that the usage of the independent coupling may result in irregular vector fields near the initial and final times. Moreover, one of the goals of the article, in contrast with previous results in the literature, is to consider much larger values of $n$. The tests carried out on datasets such as CIFAR-10 and ImageNet-32 as well as on synthetic data lead the authors to conclude that working with large Sinkhorn couplings has beneficial effects for mid-sized problem, enabling for faster training and inference. The authors also introduce a notion of "renormalized entropy" to tune the value of the regularization parameter $\varepsilon$ and recommend to keep the renormalized entropy around $0.2$

**Questions:**

- It appears to me that the usage of Sinkhorn coupling as initialization fits well with using stochastic rather than deterministic interpolants. In particular, I would use brownian bridges instead of straight lines. Have the authors thought in this direction? How do they believe their conclusion would change considering stochastic interpolants?

- I understand the general recommendation is to have a renormalized entropy around 0.2. What are the recommendations for $n$? In general, I find the plots tell little about the influence on $n$. What was the choice of $n$ for the plots in Figure $1$ and $2$?

**Ethical Concerns:**

["NO or VERY MINOR ethics concerns only"]

**Final Justification:**

I have increased the score to 5. This is because the authors addressed in a compelling way all my questions. In particular, I appreciated their clarification about the opportunity of replacing straight lines with stochastic interpolants, as well as the numerical experiments accompanying their explanation. Even though it is only a very first step, I appreciate the authors' effort to comment on the theoretical justification for their empirical findings.

**Limitations:**

yes.

**Paper Formatting Concerns:**

no concern

**Quality:**

3

**Strengths And Weaknesses:**

- The strength of the paper lies for sure in the thorough design of numerical implantation and in the extend of the numerical simulation across various datasets. Though I am not an expert in the implementation details of Sinkhorn algorithm and generative algorithms in general, I find the empirical conclusions of the article quite convincing. The benefit of working with large Sinkhorn couplings seems to be confirmed by several different metrics such as FID and curvature.


- The main weaknesses is undoubtedly the lack of any form of theoretical justification to support the experimental conclusions, though achieving this may be beyond the scope of this work

---

> ### Author Rebuttal · Authors · 2025-07-30
>
> Many thanks for your detailed comments.
>
> >**The method is meta-dataloader: training and generation coupling are separated, thus making the procedure modular and computationally realistic.**
>
> Many thanks for highlighting this point! We will do our best to underline this further in our revision.
>
> >**The main weaknesses is undoubtedly the lack of any form of theoretical justification to support the experimental conclusions, though achieving this may be beyond the scope of this work**
>
> While our main focus remains on scaling-up Sinkhorn and comprehensive experiments to study its effect, we are also able, following your comment and that of Reviewer **7qcL** to provide some theoretical intuition into the necessity of using large OT batch sizes. This result is provided as **Proposition 1** in response to **W3** of Reviewer **7qcL**.
>
> We highlight that this argument will only be shown in the appendix, as it is **only meant to provide insights into the necessity of large batch size and is not a core contribution of our work.**
>
> >**It appears to me that the usage of Sinkhorn coupling as initialization fits well with using stochastic rather than deterministic interpolants. In particular, I would use brownian bridges instead of straight lines. Have the authors thought in this direction? How do they believe their conclusion would change considering stochastic interpolants?**
>
> We agree that this is a promising direction. At the moment we do not exploit stochasticity of the coupling, but rather simply **sample** from it (Step 4) to form pairs, assuming straight paths, in accordance with the flow matching framework. Incorporating this stochasticity would be more akin to switching from ODE flow models to SDE Schrödinger bridge type models. We might explore this, or others might use our implementation to do it themselves before.
>
> In that vein, we have also tested the ability of OTFM to help training for **modality translation** (i.e. flowing from a modality to another, without having access to pairings, rather than starting from nois). This setup was typically considered by Schrödinger bridge models.
>
> We tried this with the cat→wild image translation for AFHQ-64. We follow the evaluation protocol as mentioned in [De Bortoli et al. 24]. We use their network architecture, increasing the number of channels in the U-Net to 192; train the network for 400k iterations with 512 effective batch size.
>
> As noted by Bortoli et al., visual inspection is best way to assess the quality of results. While the pictures we obtain look good, we are not allowed to share these images this year, and might want instead to add them in the appendix.
>
> As a result we can only report metrics (LPIPS, MSD and FID). As highlighted by Bortoli et al., FID is not really meaningful. as they are computed on a very small set of points (~4.5K training set of the wild domain and ~450 validation set of cat → wild transferred images). They show, however, a very clear failure of IFM vs. OTFM.
>
> These preliminary results should be compared to the values reported in Figure 19 of [De Bortoli et al. 24].
>
> |$n$|   𝜀 |   LPIPS $\uparrow$ |   MSD $\downarrow$ |     FID $\downarrow$|
> |---------:|----------:|--------:|------:|--------:|
> |      512 |     0.003 |   0.381 | 0.049 |  **23.325** |
> |      512 |     0.01  |   0.385 | 0.048 |  24.687 |
> |      512 |     0.03  |   0.383 | 0.049 |  23.496 |
> |      512 |     0.1   |   0.391 | 0.052 |  23.983 |
> |     1024 |     0.003 |   0.393 | **0.044** |  25.278 |
> |     1024 |     0.01  |   0.397 | *0.045* |  26.114 |
> |     1024 |     0.03  |   0.389 | 0.047 |  25.817 |
> |     1024 |     0.1   |   0.384 | 0.05  |  25.76  |
> |     8192 |     0.003 |   *0.407* | 0.069 |  45.939 |
> |     8192 |     0.01  |   **0.411** | 0.068 |  44.239 |
> |     8192 |     0.03  |   0.405 | 0.058 |  33.942 |
> |     8192 |     0.1   |   0.381 | 0.05  |  25.356 |
> |      IFM |     NaN   |   0.372 | 0.09  | 111.216 |
>
> [ V. De Bortoli, I. Korshunova, A. Mnih, A. Doucet, Schrödinger Bridge Flow for Unpaired Data Translation, Neurips 2024]
>
>
>
> >**I understand the general recommendation is to have a renormalized entropy around 0.2. What are the recommendations for $n$ ?**
>
>
> Our recommendation at the moment is to choose a $n$ as large as possible, within what is available in the preprocessing compute budget, since in our generation experiments show so far that larger $n$ always improves on all metrics, notably faster generation.
>
> Additionally, our implementation is now fully efficient as we have transitioned to a metaloader (See our point `3. Precomputing Noise/Data Pairs` shared with Reviewer **52g8**) in order to split entirely NN training from pairing effort, making this computation a one-off effort.
>
> >**In general, I find the plots tell little about the influence on $n$ . What was the choice of for the plots in Figure 1 and 2 ?**
>
> We apologise for not being more clear in the legends of Figs. 1 & 2 (please look at 10, 12, 16, 18 in supplementary). In all our figures, $n$ is color coded (yellow = large). We will add the mention `n=2048, n=16384, ...` in all legends. As can be seen, $n$ impacts performance in all settings.

---

> > ### Comment · Reviewer_Lgwo · 2025-08-04
> >
> > I thank the authors for addressing all my comments in detail. The answer are pretty satisfying and I will consequently increase my score.

---

> > > ### Author Response · Authors · 2025-08-04
> > > **Many thanks for acknowledging our rebuttal**
> > >
> > > We would like to thank you for taking the time to read our rebuttal. We are very happy to hear that our responses answered some of your concerns, and we are very grateful for your score increase. Above all, we thank you for helping us improve our draft!

---

### Official Review · Reviewer_7qcL · 2025-06-29

**Clarity:** 3
**Significance:** 2
**Originality:** 2
**Rating:** 4
**Confidence:** 3

**Summary:**

This paper studies at what regimes of mini-batch size $n$, regularization $\varepsilon$, and for which data dimensions $d$ could batch-OT flow matching work well. The paper interpolates Batch-OT and independent FM with the regularization $\varepsilon$ in entropic OT (EOT). The paper also modifies the Sinkhorn algorithm by dropping the square norms and focusing on the dot-product between points. Finally, the paper experiments on various tasks with substantially different regimes of $n$ and $\varepsilon$ to show the thoroughness of the proposed study.

**Questions:**

1. To balance between training time and mini-batch size $n$, is there any criterion on how to choose the proper $n$ when the total training time is limited?

2. Is it possible to give a theoretical complexity analysis like [1] to show the effectiveness of the modification of Sinkhorn algorithm?

3. In the final row of Figure 1, why are there 6 different curves when $n$ only have 5 choices? The colors and the marks of the curves do not match the legend (there are two blue lines, and the colors are mismatched).

[1] Pham, K., Le, K., Ho, N., Pham, T., & Bui, H. (2020, November). On unbalanced optimal transport: An analysis of sinkhorn algorithm. In International Conference on Machine Learning (pp. 7673-7682). PMLR.

**Ethical Concerns:**

["NO or VERY MINOR ethics concerns only"]

**Final Justification:**

I have increased the score to 4. I am delighted to see the additional theoretical analysis (the lower bound proposition) on further explaining the “necessity of large batch size” $n$, which could provide potential insights on the mathematical understanding of the proposed implementation tricks, even though a comprehensive theoretical justification is yet to be established in future works. The additional experimental detail discussions in the rebuttal of the Reviewer zUcn and the Reviewer 52g8 empirically showed the effectiveness of the implementation tricks. My other questions are also adequately addressed by the authors.

**Limitations:**

The paper addressed the limitations. It would be beneficial to add the theoretical complexity analysis that includes the sample complexity $n$. Criterions for selecting proper $n$ given a limited time budget remains to be established.

**Paper Formatting Concerns:**

No.

**Quality:**

2

**Strengths And Weaknesses:**

Strengths:
1. The paper is clearly-organized and set the central focus on the modification and influencing factors of the Sinkhorn algorithm.
2. The paper experiments on various synthetic and real benchmarks with detailed implementation to show the thoroughness of the study.

Weaknesses:
1. The contribution of the paper is limited. The main innovative point is dropping norms and focusing on the dot-product between points in the Sinkhorn algorithm for better stability, which is a narrow view to some extent. The overall contribution is mostly empirical. Some experimental details like the types of GPU could be moved to the appendix to give space to further discussions.
2. The paper states that larger batch size is needed to cover the bias that cannot be traded off with more iterations on the flow matching loss. On the other hand, Figure 1 and Figure 2 show that larger batch size suffers from longer training time.
3. It would be beneficial to include some theoretical complexity analysis like [1] that includes the sample complexity $n$ to further explain the “necessity of large batch size”.
4. Multiple independent runs of the experiments could be done to obtain the confidence interval of the results.

[1] Pham, K., Le, K., Ho, N., Pham, T., & Bui, H. (2020, November). On unbalanced optimal transport: An analysis of sinkhorn algorithm. In International Conference on Machine Learning (pp. 7673-7682). PMLR.

---

> ### Author Rebuttal · Authors · 2025-07-30
>
> Many thanks for taking the time to review our paper, and providing several comments.
>
> > **The paper is clearly-organized**
>
> Thanks!
>
> >**W1. The contribution of the paper is limited. The main innovative point is dropping norms and focusing on the dot-product between points in the Sinkhorn algorithm for better stability, which is a narrow view to some extent. The overall contribution is mostly empirical.**
>
> We agree that our contributions are driven by empirical concerns, more precisely by the goal to scaling up $n, d$ to unprecedented regimes, while tracking convergence rigorously (small $\tau$, L.213), and staying as arbitrarily close to sharp couplings (small $𝜀$).
>
> In that sense, our paper is informed by the “bitter lesson” (Rich Sutton) applied to OT computations, and is trimmed to the minimum to work in this case.
>
> While our findings may seem simple in retrospect, from **Lemma 2**, to $\mathcal{E}$, the adoption of the dot product cost in L. 190, and rescaling by `std` , we are confident that these modifications will be impactful and become the norm when running Sinkhorn.
>
> In addition to this, we have incorporated additional tricks (see  `1. Warmstart` , `2. Computing matchings in PCA Space` in our response to Reviewer **zUcn**) and demonstrated that they speed up computations significantly in our setting.
>
> >**Some experimental details like the types of GPU could be moved to the appendix**
>
> Hardware specifications only take a few characters in our draft. We prefer to be sure the reader is aware of these aspects, as they matter practically speaking, and can help get a clearer picture. Are there other details you would like to see moved?
>
> >**W2.The paper states that larger batch size is needed to cover the bias that cannot be traded off with more iterations on the flow matching loss. On the other hand, Figure 1 and Figure 2 show that larger batch size suffers from longer training time.**
>
> Indeed, using larger $n$ and decreasing 𝜀 incurs a larger compute effort. We ask, however, to consider the following aspects:
>
> * Coupling more carefully noise and images is **not**, strictly speaking, a training effort, it is a data-preprocessing effort, independent of model training (L.142) as highlighted in L. 141 and expanded in item `3. Precomputing Noise/Data Pairs` shared with Reviewer **52g8**.
> * The cost to compute such couplings is parameterised by $n, d, 𝜀$ and is independent of model parameters, which can run anywhere from billions to hundred of millions, and will always dominate overall training cost.
> * We believe, following discussions on Warmstarting and PCA, that one may continue improving on the efficiency of computing couplings.
> * A larger effort spent on better couplings is always associated in our image experiments with a higher quality when inference time compute is constrained. This is usually desirable.
>
> >**W3. It would be beneficial to include some theoretical complexity analysis like [1] that includes the sample complexity to further explain the “necessity of large batch size”.**
>
> Thanks for this great suggestion. We will gladly include [1] to the 5 references we have provided in L.145-150. We will expand this section and include any other suggestions you may have.
>
> Following your comment, we can add **(in response to your comment that this would be beneficial)** to our appendix a simple mathematical argument, similar to [Chewi et al. 2024, Thm. 2.14], that justifies the use of large batch sizes. We emphasize that this argument is **only meant to provide insights into the necessity of large batch size and is _not_ a core contribution of our work.** Using this insight to establish a tight characterization of the curvature of flow models as a function of the OT batch size would require significantly more work, which can be an interesting research direction.
> * * *
> **Assumption 1**: The data distribution $\mu_1$ is such that if $X$ and $X'$ are independently drawn from $\mu_1$ then,
> $$\mathbb{P}[\Vert X - X\' \Vert \geq t] \leq Ct^r,$$
> for all $t > 0$ and some constants $C,r > 0$.
>
> This assumption holds for example if $\mu_1$ is supported on a manifold of dimension $r$ with bounded density w.r.t. the volume measure on the manifold. Specifically, _$r$ can be seen as the effective dimension of the support of $\mu_1$, and for structured data distributions we expect $r \ll d$._
> * * *
> **Proposition 1**: Given two probability measures $\mu_0, \mu_1 \in \mathcal{P}_2(\mathbb{R}^d)$ where $\mu_1$ satisfies Assumption 1, suppose $\boldsymbol{X}_0, \boldsymbol{X}_1 \sim \mu_0^{\otimes n} \otimes \mu_1^{\otimes n}$, i.e. they represent $n$ i.i.d. samples from $\mu_0$ and $\mu_1$ respectively. Let $\hat{\pi}(\boldsymbol{X}_0,\boldsymbol{X}_1)$ denote any coupling supported on $\boldsymbol{X}_0$ and $\boldsymbol{X}_1$, including (entropic) optimal transport. Then,
> $$\mathbb{E}\_{X_0,X_1 \sim \hat{\pi}(\boldsymbol{X}_0,\boldsymbol{X}_1), \boldsymbol{X}_0, \boldsymbol{X}_1}[\operatorname{Var}(X_1 | X_0)] \geq cn^{-2/r}$$
> for some constant $c > 0$ depending on $C$ and $r$.
> * * *
> In Proposition 1, we look at the variance (sum of coordinate variances) of data conditioned on the noise it is paired to. Note that for IFM, this is simply the variance of the data distribution, as the noise/data coupling is independent. Ideally, with OT couplings we would always couple noise to a unique data sample, thus this variance would be zero and the flow trajectory would be straight. The above lower bound shows the necessity of large batch-size $n$ due to the slow rate of $n^{-2/r}$, which motivates us to scale Sinkhorn to as large $n$ as possible. While we do not include it for brevity, one can obtain matching upper bounds through standard law of large numbers arguments [Chewi et al. 2024, §2.5.1].
>
>
> >**W4. Multiple independent runs of the experiments could be done to obtain the confidence interval of the results.**
>
> Thanks for this suggestion. Unfortunately, at this time, most papers in flow matching or large scale generative modeling do not include error bars when using image generation, e.g. Table 5 in [Tong et al.], Table 1 & 2, Figure 3 in [Pooladian].This is because each run takes anywhere between 4 days to 2 weeks.
>
> To make things worse, in our case we report dozens of $n,𝜀$ variants. Producing error bars would multiply by 5 our already very substantial compute budget.
>
> Additionally, the randomness in coupling computations is **independent** from that resulting from neural network parameter initialization, increasing the challenge further (to get error bars for FM metrics, one would need to rerun multiple seeds for couplings $\times$ multiple seeds for NN training)
>
> This being said, we observe qualitatively fairly stable results, as long as we reuse similar config files. For instance, we ran the learning rate ablations (in answer to Reviewer **3ETv**) **** using different NN seeds, and obtained similar results. We sometimes observed variability in FID computations across experiments, due to the batch of 50k images that is sampled. We always ensured, however, that the multiple runs done within a single experiment use the same batch, but were not always able to guarantee this across experiments.
>
> >**Q1. To balance between training time and mini-batch size $n$ , is there any criterion on how to choose the proper $n$ when the total training time is limited?**
>
> Compute involves three independent terms:
> * pre-processing, to select pairs of noise / data  → this is where OTFM happens.
> * training time: pick FM model, optimizer, learning rate, epochs, etc.. fed with pairs from above.
> * inference time: ODE integrator, # of steps.
>
> Any rule to select $n$ must depend on how these compute budgets are prioritised, as with LLMs training.
>
> Nowadays, pre-processing steps are allocated a high budget, training budget is even higher, and inference budget is assumed to be limited. A message of our paper is that $n$ should be as large as possible (although probably not larger than dataset size, a lesson learned with the relatively toy-ish CIFAR-10 experiment).
>
> In practice, it is easy to estimate the required time to run a single Sinkhorn computation: Sinkhorn time is very stable across batches for a given $n,𝜀$ choice (see e.g. Fig. 8, top, in supplementary). The practitioner can then fairly easily calibrate their training setup with a dry run, and adjusting $n,𝜀$ according to the preprocessing compute they wish to spend, and store these results as per item `3. Precomputing Noise/Data Pairs` shared with Reviewer **52g8**.
>
> >**Q2. Is it possible to give a theoretical complexity analysis like [1] to show the effectiveness of the modification of Sinkhorn algorithm?**
>
> The modifications we provide are not theoretically motivated. For instance,
> * the `std` rule is only designed to help set 𝜀 adaptively, robustly and consistently across many setups.
> * removing norms does not change the convergence analysis of Sinkhorn, as described in Remark 4.12 [Peyré / Cuturi]. It can be interpreted as a way to bypass completely norm information in dual scalings, whereas convergence theory of Sinkhorn focuses on lower-bounding the improvement from one iteration to the other.
>
> >**Q.3 In the final row of Figure 1, why are there 6 different curves when only have 5 choices? The colors and the marks of the curves do not match the legend (there are two blue lines, and the colors are mismatched).**
>
> We apologise for this mistake and will redraw this plot. The Lowest curve can be dropped and corresponds to the setting $n=256$ that we decided to remove to improve legibility. See also our response to **W1** of Reviewer **zUcn**
>
> >It would be beneficial to add the theoretical complexity analysis that includes the sample complexity $n$. Criterions for selecting proper $n$ given a limited time budget remains to be established.
>
> Thanks, as mentioned above, we will discuss [1] and add the discussion above.

---

> > ### Comment · Reviewer_7qcL · 2025-08-05
> >
> > I thank the authors for the detailed rebuttal. I am delighted to see some mathematical argument (the lower bound proposition) on further explaining the “necessity of large batch size” $n$. I also read the additional experimental details in the rebuttal of the Reviewer zUcn and the Reviewer 52g8, which showed the effectiveness of the implementation tricks such as computing matchings in PCA space and using larger OT batchsizes. Based on the responses and the discussion I will increase my score.
> >
> > I hope the responses and revisions made in the rebuttal make their way into the paper.

---

> > > ### Author Response · Authors · 2025-08-06
> > > **Many thanks for staying available throughout the rebuttal**
> > >
> > > We would like to thank you for taking the time to read our response. Of course, we commit to including all of the elements discussed in this rebuttal process in the final draft. We are grateful for your time and consideration!

---

### Official Review · Reviewer_52g8 · 2025-07-01

**Clarity:** 2
**Significance:** 3
**Originality:** 2
**Rating:** 4
**Confidence:** 4

**Summary:**

This paper provides an ablation study of various techniques and parameter settings for coupling the samples used for training of flow-matching models. In flow-matching, samples from a reference distribution are linearly interpolated with samples from a target distribution and the resulting interpolations are used to learn a velocity field for dynamic transport between the target and the reference. In most implementations of flow-matching, the samples from the target and reference are coupled independently to create the endpoints of the interpolations, but recent work has suggested that it may be beneficial to couple the points using optimal transport (OT), or approximations thereof, in order to stabilize training and decrease the computational burden of inference. In practice, points from the target and reference can be coupled using "minibatch optimal transport," wherein subsets of $n$ points are coupled to each other using, e.g, the Sinkhorn algorithm. This paper experiments with using couplings where $n$ is taken much larger than the oft-used $n = 256$, obtained using the Sinkhorn algorithm with various regularization levels $\varepsilon$. The paper also suggests some practical guidelines for choosing $\varepsilon$, computing coupling entropies, and preparing data for computation of large couplings. The numerical results generally suggest that using large couplings with low regularization $\varepsilon$ is beneficial in comparison to using the independent coupling.

**Questions:**

* As mentioned under "Weaknesses," the $y$-axes of the plots used to display performance metrics as a function of $n$ and $\varepsilon$ all need to start at 0 so as not to suggest that significant trends are present when they are really not. Please fix these axes in your next revision of the manuscript.

	 Even though it is perhaps disappointing that the benefit of increasing $n$ and decreasing $\varepsilon$ for many of the experiments was unclear, if you were to run more experiments and report trends (or lack thereof) in a clear way, I wonder if a pattern would emerge that would hint at a distinction between types of problems that benefit strongly from couplings with large $n$ and low $\varepsilon$, and those that don't? As far as I can tell, the only example where such benefit was clearly and strongly evident was the Korotin et al. benchmark. What makes the Korotin et al. benchmark different from the other examples you tried? If you were to run more experiments along these lines, would you be able to hypothesize what it is about the Korotin et al. benchmark and other examples where large low-entropy couplings are beneficial, that differentiates them from examples where these couplings are not beneficial? From a practical standpoint, even being able to say "this type of problem doesn't benefit from large couplings, so don't bother deviating from IFM" or "this problem does benefit from large couplings, so put in the effort to get one before you train" could go a long way to inform effective Flow-Matching practice.

* What is the rationale for scaling the entropy level $\varepsilon$ by the standard deviation of the cost-matrix $\bf C$, rather than the median or mean, as in [Cuturi 2013]? The stated reasoning -- that "the dispersion of the costs around its mean has more relevance than the mean itself" -- doesn't clarify for me why using the standard deviation is a good practical choice.  Do you have numerical results comparing this choice to, e.g., scaling by the mean, that demonstrate clear benefit? Is there a deeper mathematical reason for why you made this choice?

* There are many instances in the paper where the mathematical language could be more precise. Removing mathematical ambiguities would go a long way toward making the paper more clear and readable. For instance:
	* In equation (1) on page 2, the Wasserstein distance denoted $W(\mu, \nu)$ is specifically the Wasserstein-2 distance (or a scalar multiple thereof). Consider changing notation to $W_2(\mu, \nu)$ .

	* On lines 115-117, it is stated that "as $\varepsilon \to 0$, the solution $\mathbf P^\varepsilon$ converges to the optimal transport matrix solving (1)" -- Equation (1) is the *continuum* optimal transport problem, and therefore its solution is a *coupling*, not a matrix. Its solution could, however, be viewed as a matrix if the measures $\mu$ and $\nu$ are taken to be discrete/empirical measures. Please clarify/correct this statement.

	* On line 126 the random variable $T$ in the flow-matching loss is described simply as "a random variable in $[0,1]$". Shouldn't $T$ specifically be uniform over $[0,1]$?
	* On line 169, it is stated that the renormalized entropy $\mathcal E$ provides a measure of how close $\mathbf P^\varepsilon$ is to an optimal assignment matrix. Wouldn't it be more accurate to say that $\mathcal E$ provides a measure of how close $\mathbf P^\varepsilon$ is to *any* deterministic assignment matrix? The coupling does not have to be optimal to have zero entropy. That is, $\mathcal E$ does not actually tell us whether we are close to the optimal assignment, just whether we are close to any 1-1 assignment.

**Ethical Concerns:**

["NO or VERY MINOR ethics concerns only"]

**Final Justification:**

While the initial manuscript had some serious clarity issues and included methodological contributions that were marginal at best, based on discussion with the authors I feel that the clarity of the final manuscript will improve. Moreover, the authors have devised more computational strategies which, when combined with the existing rules-of-thumb in the first submission, provide a more comprehensive toolbox for computing large couplings using the Sinkhorn algorithm. These factors, combined with the authors' results which indicate that using large couplings to obtain flow-matching training pairs really does seem to improve performance, lead me to believe that the contributions of this paper have the potential to inform future flow-matching practice.

**Limitations:**

Yes

**Paper Formatting Concerns:**

The citation styles used are inconsistent and out-of-step with NeurIPS guidelines. Specifically, multiple citations are included as hyperlinks of an author's name with no year (e.g., "Monge", "Benamou and Brenier", "Sinkhorn"). Please ensure that your citation style is consistent and agrees with NeurIPS guidelines (i.e., is either author-year or numeric). Many of these hyperlinks could probably just be removed (e.g., there's no need to cite Sinkhorn every time you refer to the "Sinkhorn algorithm").

**Quality:**

3

**Strengths And Weaknesses:**

## Strengths
* The paper computes the data couplings over multiple GPUs and provides details on how the data are distributed, sharded, and regathered. I am not well-acquainted with the SOTA in GPU computing for Sinkhorn/OT, but, according to the authors, GPU computations of this scale for obtaining Sinkhorn couplings have never been reported in the literature.

* For the image generation benchmarks, the authors show flow-matching results not only for various $(n, \varepsilon)$ but for different ODE solvers used in generation.  Choice of ODE solver is an important practical consideration when implementing flow-matching, and having some insight on how that choice interacts with other hyperparameters is useful. In particular, using Dopri5 instead of forward Euler with low numbers of steps seemed to remove the impact of the couplings to a large extent. This is a salient finding -- if one does not want to tune coupling hyperparameters in flow-matching, one can just use a better ODE solver for generation.

## Weaknesses
* The paper is poorly written and the wordings used are frequently ambiguous, unclear, or perhaps even mathematically incorrect. As a result, I found the paper difficult to parse in spite of the facts that the contributions are not too complicated and I am well-versed in flow-matching and optimal transport.

* The methodological contributions of the paper are marginal at best.  For example,
	* One stated contribution of the paper (on page 2) is "Leveraging the fact that all [OT-based coupling] approaches can be interpolated using EOT: Hungarian [exact OT] corresponds to the case where $\varepsilon \to 0$, while IFM [using the independent coupling] is recovered with $\varepsilon \to \infty$" -- I think this continuum is fairly widely understood.

	* The automatic rescaling of $\varepsilon$ amounts to multiplying a scale-free $\varepsilon$ by the standard deviation of the entries of the cost matrix and is not very different from the scaling used in [Cuturi 2013], wherein $\varepsilon$ was multiplied by the median of the distances.  Moreover, the rationale for using the standard deviation is unclear from the paper.

	* Similarly, the scale-free renormalized coupling entropy amounts of the usual entropy divided by $\log n - 1$. It is definitely important to use scale-free quantities when comparing results across numbers of points $n$, but using scale-free quantities is standard practice in most areas of applied mathematics where dimension/number of points is varied.

* Many of the plots containing the numerical results are presented in a (I hope unintentionally) deceiving manner. As alluded to in the summary, the $y$-axes on the problematic plots do not start at zero and actually contain a very narrow range of values. If one does not look at the $y$-axis limits, it appears that there are strong trends in the performance metrics as functions of $\varepsilon$ and $n$, but these trends would not be significant were the $y$-axes started at zero. Figures which contain plots with massively truncated $y$-axes include Figure 1 (and Figure 11 in the supplemental, especially for $\mathcal R(\theta)$ and BPD), Figure 3 (almost all plots), and Figures 16, 18, and 20 in the supplemental (almost all plots).
* The numerical results for most of the benchmarks presented do not paint a clear picture of whether using large $n$ and small $\varepsilon$ is helpful, especially in high-dimensional sampling tasks. The exception is perhaps the Korotin et al. benchmark.

---

> ### Author Rebuttal · Authors · 2025-07-28
>
> Many thanks for your detailed review and your insights.
>
> We also found a few unfortunate typos (residual $m$ cardinalities in Alg.1, wrong transpose on $\mathbf{Y}$ in L.185, also missing a "+", etc...), and apologize for this. We do, however, stand by the maths in our paper.
>
> >**various OT solvers[…] including the Hungarian algorithm**
>
> Let us clarify that we never use the Hungarian algorithm, we only use Sinkhorn.
>
> >**S1. according to the authors, GPU computations of this scale[…]have never been reported in the literature.**
>
> Thanks for raising this point. Indeed, our computations are unprecedented for two reasons:
>
> **Scale.** The hardness of an OT problem hinges on many parameters, typically: $d$ (dimension); $n$ (# of points); constraint tolerance (e.g. $\tau$ in Alg. 1); intended closeness to the optimal assignment/cost (e.g. 𝜀).
>
> To our knowledge, no paper has pushed computations of Sinkhorn to this extent, e.g. for ImageNet-32, $n=524288, d=3072$ while tracking precisely constraints ($\tau= 10^{-3}$ in 1-norm) and sharpness (small 𝜀). The largest setup we know of is the computation of a few coupling ($n=60k, d=1k$) in [Kassraie & al. 2024, Fig.6].
>
> While low-rank approximate OT solvers have been used recently to compute fairly large couplings (see [Halmos, 2025], [Klein & al. 2025]), these reformulations are not convex, and offer no guarantees to solve the OT problem. Using them in large scale OT-FM could be an interesting follow-up work.
>
> **Repeats.** We compute large couplings *many* times: e.g. for ImageNet-32, with $n=524288$ points, we do so `~850` times, to cover 430k training steps.
>
> _P. Kassraie & al., Progressive Entropic Optimal Transport Solvers, Neurips 2024_
>
> _Halmos & al., Hierarchical Refinement: Optimal Transport to Infinity and Beyond, ICML 2025_
>
> >**This is a salient finding — if one does not want to tune coupling hyperparameters in flow-matching, one can just use a better ODE solver for generation.**
>
> This finding is of little use in the context of FM/diffusion.
>
> * The number of ODE steps is an **inference-time parameter**: in practice, for an end-user, better quality = more steps = longer wait.
> * Computing larger/better couplings is a **preprocessing** effort.
>
> Because inference and preprocessing are done by different actors on different hardware (e.g. consumer devices/lower grade GPU vs. GPU servers with interconnect), one cannot easily trade off one for the other. When models are shared with the public, inference cost will always dominate training cost. This is why we need fast flowing models.
>
> >**One stated contribution […] this continuum is fairly widely understood.**
>
> We agree, this continuum is well understood in OT papers, but we stress
> - in L.69 that it has not been leveraged enough in the **OT-FM literature**.
> - Pooladian / Tong have not ablated 𝜀 in their studies (L.70) and emphasized instead a distinction between Hungarian / EOT / IFM.
> - In L.73, we claim that these methods can be unified by varying 𝜀 in Sinkhorn, as long as convergence 𝝉/max_iters is rigorously tracked (L.213)
> - In L.79 we facilitate this comparison across tasks and scales using $\mathcal{E}$.
>
> > **many of the apparent trends reported on plots […] are actually marginal-to-nonexistent when one examines the y-axis limits.**
>
> > **numerical results are presented in a (I hope unintentionally) deceiving manner […] the y-axes on the problematic plots do not start at zero[…] trends would not be significant were the y-axes started at zero.**
>
> > **all need to start at 0[…] Please fix**
>
> We acknowledge your concern that FID or BPD _sometimes_ vary in a narrow y-range in our plots, but this is usual in the diffusion/FM literature. Adding a 0 would be highly unusual and clutter plots. Instead we follow the practice of very highly cited papers in the literature (and more generally ML) to let matplotlib choose the y-axis bounds:
>
> * Lipman & al, Flow Matching for Generative Modelling **FID: Fig. 5 & 7**
> * Song & al., Score-based Generative Modeling through Stochastic Differential Equations **FID: Fig. 10**
> * Liu & al., Flow Straight and Fast: Learning to Generate and Transfer Data with Rectified Flow **FID: Fig. 8; Straightness (our 𝜅): Fig. 9**
> * Nichol & al., Improved Denoising Diffusion Probabilistic Models **NLL/BPD: Fig. 15 & 16; FID: Fig. 17**
>
> While adding a 0 in our y-axes would be problematic, we commit, however, to
> - provide more tables (as done in this rebuttal),
> - highlight in the captions the narrow BPD or $\mathcal{R}$ ranges
> - remove Fig. 11 & 13, as they just present the data in Fig. 1 & 2 without the 🔻 IFM / green▲ ground-truth lower bound.
>
> >**The numerical results […] do not paint a clear picture […]exception is perhaps the Korotin benchmark.**
>
> For the **Piecewise quadratic** benchmark (Fig. 1)
> * very large $n$ (1M, 2M) improves on small $n$ (16k) and IFM on _all_ metrics.
> * The curvature 𝜅 for large $n$ is `25x` to `4x` times smaller than IFM
> * The reconstruction loss $\mathcal{R}$ of large $n$ OTFM is always `10~20%` better than IFM.
> * BPDs are comparable, but this is usual: BPD values tend to cluster when comparing similar methods (see e.g. Table 7 in [Papamakarios & al. Masked Autoregressive Flow for Density Estimation])
>
> We agree that **CIFAR-10** results lack significance, because CIFAR-10 is too small (the 50k database size is *smaller* than our larger batch sizes $n$). We will move CIFAR-10 to the appendix ([Pooladian et al.] skipped it altogether).
>
> For **ImageNet-32** and **ImageNet-64**, _all_ FID metrics for small NFE are significantly better than IFM, and 𝜅 improves as well (Figs. 16 & 18, Tables 1 & 2). This is visually striking in Fig. 17 and Fig. 19.
>
> > **Even though it is perhaps disappointing […] inform effective FM practice.**
>
> Given your lowest possible rating of 1 to the significance & originality of our work, we felt encouraged to see that it still managed to trigger so many open questions.
>
> Our findings are, however, clear: using large $n$ / small 𝜀 Sinkhorn couplings _always_ improved (or left unchanged) FM metrics in _all_ problems we considered.
>
> The magnitude of these gains depends on data, as you mention, but also on model capacity & training method. Predicting the magnitude of these gains will necessitate a "scaling laws" approach that we leave for follow-up work.
>
> >**What is the rationale for scaling the entropy level by the std of the cost-matrix[…] numerical results comparing this choice[…]**
>
> As an example, consider a $n\times n$ cost matrix $C_1$ with values distributed in $[0,1]$. Consider the shifted cost $C_2 = C_1 + 100$.
>
> * Mathematically, the EOT solution for $C_1$ is the same as that for $C_2$. Indeed, because $P$ is a bistochastic matrix, $$ \arg\min_P\langle P, C_2 = C_1 + 100\rangle - 𝜀H(P) = \arg\min_P \langle P, C_1 \rangle + 100 - 𝜀H(P) = \arg\min_P\langle P, C_1 \rangle - 𝜀H(P).$$
> * Hence, EOT solutions are **invariant to the addition of a constant in the cost when using the same 𝜀** (a particular case of L.187 with a _global_ shift)
> * Hence, a data dependent rule to set 𝜀 (or its scale) should retain this shift-invariant property.
> * The `mean,` `median` or `max` rules introduced in [Cuturi ’13] are _not_ shift-invariant. They were introduced to avoid underflow in the kernel $K=\exp(-C/𝜀)$. Most solvers used nowadays avoid this problem by relying on log-sum-exp computations (L.109).
> * Yet, the 𝜀 selected with `mean`, `median` or `max` rules would be `100x` larger for $C_2$ compared to $C_1$, despite the fact that these problems are highly similar.
>
> To scale 𝜀 in a shift-invariant way, we propose to quantify **dispersion** with `std`. While `std` is not the only way to achieve this, it is cheap and robust.
>
> We do observe qualitatively that `std` scaling results in lower variance of # Sinkhorn iterations / renormalised entropy across experiments, compared to `mean`. We will clarify in the appendix.
>
> >**changing notation to W_2**
>
> We will remove the Wasserstein notation entirely, and only introduce OT couplings.
>
> >**On lines 115-117 […] Eq. (1) is the *continuum* OT problem, and therefore its solution is a *coupling*, not a matrix.[…]**
>
> Initially, we instantiated Eq. (1) with $\mu_n=\tfrac1n\sum_i \delta_{x_i}$ and $\nu_n=\tfrac1n\sum_j \delta_{y_j}$. We removed this when shortening the paper. We will revert and clarify.
>
> >**the random variable $T$[…] Shouldn't specifically be uniform[…]?**
>
> Time is not necessarily sampled uniformly, see e.g. paragraph above Eq. 4.27 in [Flow Matching Guide and Code]. We will clarify.
>
> >**On line 169, it is stated […] $\mathcal{E}$ does not actually tell us whether we are close to the optimal assignment, just whether we are close to any 1-1 assignment.**
>
> Thanks for the opportunity to clarify.
>
> While we define $\mathcal{E}(P)$ for *_any_* coupling (L. 167), we write in L.168 *“$\mathcal{E}(\mathbf{P}^𝜀)$ provides a simple measure of the proximity of $\mathbf{P}^𝜀$ to an optimal assignment matrix”*. **We evaluate $\mathcal{E}$ exclusively on Sinkhorn EOT solutions $\mathbf{P}^𝜀$ in the regularization path.**
>
> On that path:
> * $\mathbf{P}^𝜀$ interpolates between $\mathbf{P}^\star$ (the optimal permutation, assuming it is unique w.h.p.) and $\mathbf{1}_{n\times n}/n^2$.
> * $H(\mathbf{P}^𝜀)$ strictly increases as 𝜀 grows (e.g. Eq. 4.5 in [Peyré and Cuturi’19]), and is s.t. $\log(n)\leq H(\mathbf{P}^𝜀)\leq 2 \log n$.
> * Therefore, if $\mathcal{E}(\mathbf{P}^𝜀)\approx 0 \Rightarrow H(\mathbf{P}^𝜀) \approx H(\mathbf{P}^\star) \Rightarrow\mathbf{P}^𝜀 \approx \mathbf{P}^\star$.
>
> Hence, qualitatively speaking, $\mathcal{E}$ is a cheap proxy to quantify closeness of $\mathbf{P}^𝜀$ to $\mathbf{P}^\star$ without having to compute $\mathbf{P}^\star$ itself.
>
> >**citation styles used are inconsistent and out-of-step with NeurIPS guidelines.[…]**
>
> We follow the Neurips guidelines (Section 4.1) and use the natbib `\citeauthor` macro to highlight famous names/results, to facilitate reading. We can remove this.

---

> > ### Comment · Reviewer_52g8 · 2025-08-02
> >
> > I thank the authors for their detailed response.
> >
> > - Thank you for clarifying that you do not use the Hungarian algorithm; what I was alluding to is that in the piecewise affine and Korotin benchmarks you have access to a ground-truth OT coupling and use it as a basis of comparison to the Sinkhorn OT couplings. I will update my summary.
> >
> >
> > - With regard to the axis limits of the plots, while I agree that it is not necessary to start the y-axis exactly at zero on every single plot, it *is* necessary to include a significant range of y-values on the axis so that the reader can easily judge the magnitude of the trend in the data.  The extremely narrow y-limits resulting from the use of matplotlib defaults are misleading because the resulting plots suggest that dramatic trends exist when they in fact do not. For instance, the limits on the y-axis on the leftmost “BPD” plot in Figure 1 are (2.765, 2.775) — meaning that the upper limit differs from the lower limit by less than 1%.  By contrast, the plots that the authors cite to justify the use of the matplotlib defaults all contain y-axis limits which differ by at least 100% (with the exception of the straightness plot in the rectified flow paper, which I would also argue does not demonstrate best practice). Thus, suggesting that that the plots I flagged as problematic are in line with plots in highly cited ML papers isn’t credible.  Presenting the data in question in tabular form would be a good alternate option for the experiments where the metrics varied in a very narrow range, but if you do not want to widen the y-axes on the problematic plots I think it would be best to remove the plots entirely.  This is a matter of clarity and credibility: we want it to be easy for readers to accurately assess the results of your experiments. Including plots with very narrow y-ranges will confuse readers, cause readers to think that you are intentionally trying to deceive them, or both.
> >
> > - Regarding the improvement in FID on ImageNet, I see that using a large OT coupling results in marked improvement over the independent coupling when integration is performed with Euler. However, the differentiation between various settings of $n$ and $\epsilon$ seems small to me, especially for ImageNet-64. From a practical standpoint, will small decreases in FID resulting from increased $n$ be noticeable in the generated images? Figures 17 & 19 don’t show strong improvements with increased batch size to my eye. Maybe there are smaller values of $n$ that one could use and still obtain similar images? I.e., could you identify a “threshold” value of $n$ that resulted in acceptable/saturated image quality? I see that you are running some follow-up experiments with smaller $n$ in response to reviewer **zUcn**, which should help elucidate this threshold. As you mentioned, saturation of image quality with increasing $n$ could be an interesting question to address theoretically in follow-up work.
> >
> > - Thank you for the clarification about scaling by the standard deviation; I see the rationale now. If you have room, I would suggest adding some of these clarifying comments in the main body of the paper, since the choice of $\varepsilon$ is one of the main stated contributions of the paper.

---

> > > ### Author Response · Authors · 2025-08-02
> > > **Thank you for taking the time to read our rebuttal.**
> > >
> > > > Thank you for clarifying that you do not use the Hungarian algorithm [...] I was alluding to is that in the piecewise affine and Korotin benchmarks you have access to a ground-truth OT coupling and use it as a basis of comparison to the Sinkhorn OT couplings.
> > >
> > > Thanks! You are in fact alluding to an **extremely** important point, that is often missed in our opinion, when using a ground truth OT map to benchmark OT / flow solvers.
> > >
> > > The most important difference when training with a ground truth OT baseline is **not** that we provide the ground truth coupling to the ground truth approach, but rather in the way training batches are constructed :
> > > - For ground truth OT, in the same training batch, every ground truth origin point $\mathbf{x}_i$ is associated to its corresponding target $T_0(\mathbf{x}_i)$, see L. 258.
> > > - For our method and IFM, we feed independently generated batches of source and target, i.e. $\mathbf{x}_i$ and $T_0(\mathbf{x}'_i)$ L.254: all of the subtlety lies in **using independently resampled source points**, $\mathbf{x}'_i$. In other words, even "disambiguating" an unpaired batch by running an exact OT coupling on such batches  does not provide paired source and corresponding "true" target, since the exact OT transport $T_0(\mathbf{x}_i)$ of a source point is _never_ in the target set. This approach makes it, naturally, much harder for estimators to handle high-dimensions, but this is the only realistic setting to benchmark OT approaches.
> > >
> > > > With regard to the axis limits of the plots, while I agree [...], it is necessary to include a significant range of y-values on the axis so that the reader can easily judge the magnitude of the trend in the data. [...] This is a matter of clarity and credibility [...]
> > >
> > > We understand and acknowledge your concern. In our opinion, the clearest "offender" to your point is clearly our BPD plots. We commit to moving BPD results to the appendix (this will free space for material following you and other reviewers recommendations) and only provide them as a table, removing that part of the plot.
> > >
> > > We feel that the other metrics that matter most ($\mathcal{R}$, 𝜅, FID) are fairly represented in their y-axis. If anything, we even sometimes preferred understating the much better performance of our methods vs. IFM 🔻w.r.t **curvature** 𝜅 in the Piecewise quadratic benchmark, by using arrows/numerical values for IFM 🔻 (top of Figure 1 / 10). We felt that a comparison to the ground truth lower-bound was more useful there.
> > >
> > > > [...] I see that using a large OT coupling results in marked improvement over the independent coupling when integration is performed with Euler. However, the differentiation between various settings of $n$ and $\epsilon$ seems small to me, especially for ImageNet-64.
> > >
> > > Maybe there is a bit of information overload, as we targeted too many results across $n$ and 𝜀. In that regard Table 1 and Table 2 may be easier to interpret, as they use a single 𝜀, and drive more simply the "large $n$ is better" message.
> > >
> > > > From a practical standpoint, will small decreases in FID resulting from increased $n$ be noticeable in the generated images? Figures 17 & 19 don’t show strong improvements with increased batch size to my eye.
> > >
> > > While this is subjective, we would argue that there is a gradual improvement in image quality as one grows larger $n$ (they look sharper) for the leftmost columns. Ultimately, this is reflected in the smaller FIDs that we see for Euler integration.
> > >
> > > > Maybe there are smaller values of $n$ that one could use and still obtain similar images? I.e., could you identify a “threshold” value of $n$ that resulted in acceptable/saturated image quality?
> > >
> > > Measuring this "acceptable" quality is a difficult problem... To discuss such improvements, we see no other alternative than FID at this point, for lack of a better measuring stick. We used other accepted metrics in the cat→wild modality translation experiment presented to Reviewer **Lgwo**.
> > >
> > > > I see that you are running some follow-up experiments with smaller $n$ in response to reviewer zUcn
> > >
> > > Our point to reviewer **zUcn** was mostly about re-adding the poor to very poor results that we saw at $n=256$, and committed to adding them (see also our latest answer above). Other than that, we think that the trend, if one digests the wide variety of $n/𝜀$ settings, point very clearly to improving results with larger $n$ (e.g. Table 1 / 2)
> > >
> > > > Thank you for the clarification about scaling by the standard deviation; [...] suggest adding some of these clarifying comments in the main body of the paper, since the choice of $\varepsilon$ is one of the main stated contributions of the paper.
> > >
> > > Definitely! It is absolutely our idea to add our answers to all of the comments you and other reviewers have provided so far to improve the readability of this draft.
> > >
> > > In that sense, we are grateful for your comments that have helped improve our work, and thank you again for your reviewing time.

---

> ### Author Response · Authors · 2025-08-03
> **Another addition to your question on small $n$**
>
> Following the comment of Reviewer **zUcn** on small OT batch size $n$, we had committed to include small $n$ experimental results in the draft. This is also a topic that appeared in your latest response to our rebuttal, hence we take this opportunity to share our latest results.
>
> We are happy to report preliminary experiments with small $n=256$ on ImageNet-32, which confirm that while using small OT batch sizes $n=256$ can offer _some_ FID gains over IFM for small NFE, it may also perform worse relative to IFM for larger NFE/Adaptive solvers. On the other hand $n=256$ is always significantly worse than large $n$.
>
> The following table shows our experiment with small OT batch size $n=256$ (i.e. $32$ images per device) on ImageNet-32, and compares it with IFM and a larger OT batch size. As seen from the table,
>
> * large OT batch size $n= 524288 $ is substantially better than small $n=256$ for all metrics.
> * For large NFE, we even see that IFM edges above $n=256$ regardless of 𝜀.
>
> | $n$ OT Batch Size | 𝜀 | FID@NFE=4 | FID@NFE=8 | FID@NFE=16 | FID@Dopri5 | Checkpoint (steps) |
> |----|----- |----|-----|----|----|---|
> | IFM           | NA      | 65.8      | 23.9      | _12.1_     | _5.38_     | 180K               |
> | 256           | 0.3     | 42.1      | 20.2      | 13.8       | 8.48       | 180K               |
> | 256           | 0.1     | _39.7_    | _19.4_    | 13.4       | 8.16       | 180K               |
> | 256           | 0.03    | _39.7_    | 19.7      | 13.4       | 8.15       | 180K               |
> | 524288        | 0.03    | **30.2**  | **14.9**  | **9.54**   | **5.18**   | 180K               |
>
> These comparison are made at the 180K step checkpoint (out of total $\approx 430K$ schedule) because our runs with $n=256$ batch size haven’t finished yet. We note, however, that measures like FID@NFE=4/8 have already plateaued and will not improve at later checkpoints (see also our response to Reviewer **3ETv** showing that ImageNet32 metrics tend to plateau from 150k iterations, and might even slightly increase again after ~250k iterations)
>
> We are not able to show generated images (as in Fig. 17) per the rebuttal policy this year, but these images come as expected (notably blurrier at $n=256$ for small NFE compared to large $n$).
>
> Many thanks again to you (and Reviewer **zUcn**) for suggesting this experiment, we agree that it completes our message by also looking at what happens at the "left end of the range", not just at very large $n$.

---

> > ### Comment · Reviewer_52g8 · 2025-08-03
> > **Thank you for the clarifications**
> >
> > Many thanks to the authors for their detailed responses and for following up with new experimental results. I have enjoyed and learned from the discussion. Many of my concerns have been addressed and I think the manuscript will improve based on the items discussed above, so I will increase my score.

---

> > > ### Author Response · Authors · 2025-08-04
> > > **Many thanks for taking the time to engage in this rebuttal**
> > >
> > > We are very grateful for your time and engagement during this conversation, and we are very thankful for your score increase. We are in particular very happy to hear that we have addressed your concerns. We commit to including our answers above to the draft. Many thanks again for helping us improve our draft!

---

### Official Review · Reviewer_zUcn · 2025-07-02

**Clarity:** 4
**Significance:** 3
**Originality:** 3
**Rating:** 5
**Confidence:** 4

**Summary:**

The authors study the problem of using mini-batch optimal transport to improve the generation capabilities of flow matching models. The authors make use of entropy regularized OT to interpolate between optimal couplings (mini-batch OT) and independent couplings (as used in practice). They leverage the setup of [1] to keep entropy regularization parameter $\epsilon$ in the range $[0,1]$ instead of $[0,\infty)$, which allows for interpretable results. They then study the effects of $\epsilon$ across different batch sizes and datasets. Their results find that using mini-batch OT can improve the generation results, especially when the batch size is large.


[1] Cuturi, Marco. "Sinkhorn distances: Lightspeed computation of optimal transport." Advances in neural information processing systems 26 (2013).

**Questions:**

- Could the authors confirm if Imagenet was done in an unconditional setup?
- The authors mentioned that the results for the main paper were not complete, could you please provide the new results?
- See weakness section

**Ethical Concerns:**

["NO or VERY MINOR ethics concerns only"]

**Final Justification:**

The use of mini-batch OT as a coupling to train flow models is a technique that was not completely well understood. This paper sheds light on to when should one use mini-batch OT. Initially they were lacking some results, but after the discussion they have agreed to include them. I believe that after adding those it paints a complete picture of the roll of mini-batch OT.

**Limitations:**

yes

**Paper Formatting Concerns:**

-

**Quality:**

4

**Strengths And Weaknesses:**

Strengths

- The paper is very well written, with a nice introduction to the topics and literature review
- The experiments set up are comprehensive, including synthetic datasets where the ground truth is available, as well as more realistic datasets like imagenet

Weaknesses

- The authors considered a minimum batch size of 2048, it would be important to know the effect of mini-batch OT in the range below that, as it is common in practice to use smaller batch sizes
- The results seem to be done in an unconditional setup, the conditional set up is however, of great importance to the community. Would the authors be able to comment/provide evidence of how these results change when considering, for instance, class labels? It seems like the task of providing the coupling can be a little trickier there, which may limit the applicability to higher scale problems

---

> ### Author Rebuttal · Authors · 2025-07-29
>
> Many thanks for your encouragements and detailed review.
>
> >**They leverage the setup of [1] to keep entropy regularization parameter $\epsilon$ in the range [0,1] instead of $[0, \infty]$**
>
> Let us clarify this:
>
> * The renormalised entropy $\mathcal{E}(\mathbf{P}^𝜀)$ is a *metric* for EOT solutions taking values in $[0,1]$ (L.176). This metric assesses whether the solution $\mathbf{P}^𝜀$ lies close to the optimal assignment (when $\approx 0_+$) or the independent coupling (when $\approx 1_-$), see also our answer to **Rev. 52g8**.
> * We still need to set 𝜀: we set it as a multiple of the `std` of the cost (L. 166). Choosing that multiple in $[0.003, 0.3]$ we got a sufficiently wide coverage of $\mathcal{E}$ in $[0,1]$ in experiments.
>
> >**The paper is very well written, with a nice introduction to the topics and literature review**
>
> Thanks!
>
> >**W1. The authors considered a minimum batch size of 2048, it would be important to know the effect of mini-batch OT in the range below that […]**
>
> This is a great point.
>
> We used $n=256$ early on in the piecewise affine benchmark and observed fairly poor results for $\mathcal{R}$ and 𝜅 (we were not tracking BPD at that time), see table below for $d=32/256$, which should be compared with other $n$'s in the left/right columns of Fig. 1 (preferably retrieved as Fig. 10 in supplementary)
>
> | $d=32$ | 𝜀 = 0.003 | 𝜀 = 0.01 | 𝜀 = 0.03 | 𝜀 = 0.1| 𝜀 = 0.3|
> | --:| --:| --:| --:| --:| --:|
> | $\mathcal{R}$(𝜃)| 2.69| 2.67 | 2.72 | 2.80 | 3.01 |
> | 𝜅(𝜃)| 0.25| 0.25 | 0.25 | 0.27 | 0.43 |
>
> | $d=256$| 𝜀 = 0.003 | 𝜀 = 0.01 | 𝜀 = 0.03 | 𝜀 = 0.1 | 𝜀 = 0.3 |
> | --:| --:| --:| --:| --:| --:|
> | $\mathcal{R}$(𝜃)| 24.83| 24.99 | 25.17 | 25.06| 25.48|
> | 𝜅(𝜃)| 0.51 | 0.50| 0.50| 0.52 | 0.60 |
>
> Performance drops compared to $n\geq 2048$. More surprisingly, the reconstruction metric $\mathcal{R}$ for $d=256$ ($\approx 25$) is much worse than IFM ($\approx 21.6$) or OTFM baselines ($\leq \approx 21.4$).
>
> * for **small 𝜀**, we hypothetize that this is due to the statistical bias of exact OT for large $d$ / small $n$.
> * For **large 𝜀**, one would expect results to be, in principle, more similar to IFM (they both result in independent sampling). This difference is likely due to the way we implemented Alg. 1, more specifically Step 3.
>   * When using IFM, we set $\mathbf{P}=\mathbf{I}_n/n$ (L.136), i.e. the arbitrary identity assignment;
>   * For OTFM we use $\mathbf{P}^𝜀$, which looks like $\mathbf{1}_{n \times n}/n^2$ as 𝜀 grows.
>   * Both procedures are roughly equivalent for large $n$ but differ for very small $n$ because OTFM would be less sample efficient, as it picks samples with replacement and likely duplicates some data points while dropping others (stratified vs. non-stratified sampling).
>
> Generally, using very small $n$ was challenging in our earlier implementation, as it resulted in coupling sizes that were potentially smaller than the default batch size needed to train FM models.
>
> We will rerun some small $n$ experiments for ImageNet32/64 and add this discussion to the appendix.
>
> > **W2. Could the authors confirm if Imagenet was done in an unconditional setup?**
>
> Yes, all results in the paper & supplementary were **unconditional**.
>
> >**W2. […] conditional set up is however, of great importance to the community. Would the authors be able to comment/provide evidence of how these results change when considering, for instance, class labels?**
>
> [Chemseddine 2023] and others have applied OT-FM to the class-conditional setup, augmenting the feature space with a one-hot class label. We extended our ImageNet32 experiments to conditioned generation over 1000 classes. The following FID results are for 𝜀=0.1/I-FM after 180k steps, and renormalized entropy is consistently ~0.1.
>
> ### Imagenet 32: conditional generation FID
> ||Dopri5|Euler4|
> |:-|-:|-:|
> |I-FM|3.68|32.37|
> |$n=2048$|4.17|29.37|
> |$n=16384$|3.74|26.06|
> |$n=65536$|3.52|24.79|
> |$n=262144$|**3.47**|**23.68**|
>
> When testing conditional generation, we plan to move to latent-space representations. We prioritized for now unconditional / pixel space generation because the community wants to see generation methods operate reasonably well in pixel space (performance is harder to judge visually when rendered using high-quality latent spaces).
>
> _[ Chemseddine J, et al.,Conditional Wasserstein distances with applications in Bayesian OT flow matching ]_
>
> >**The authors mentioned that the results for the main paper were not complete, could you please provide the new results?**
>
> Indeed, the main paper results were incomplete at time of submission, as some of our jobs (e.g. Imagenet32/64) took a few days longer than expected. We apologise for this.
>
> The complete results **were submitted in the supplementary file**, 1 week after the main paper deadline. They are in the `.zip` file at the top of this page.
>
> -----
>
> # Speeding up Sinkhorn
>
> Since submission, we have progressed on our agenda to scale up Sinkhorn for OTFM to unprecendented scales. We report on the following changes, that we plan to add to our draft.
>
> ## 1. Warmstart
>
> We leverage warmstarting for Sinkhorn. Rather than always reinitialize $\mathbf{f}, \mathbf{g}$ as $\mathbf{0}_n$ in L.1 in Alg.1, we extrapolate the solution obtained when coupling the previous batches of (noise, data) to initialize $\mathbf{f}, \mathbf{g}$ for the next batches.
>
> More precisely, for a pair of optimal $n$-dimensional vectors $\mathbf{f}, \mathbf{g}$ outputted at Line 7 of Alg.1, corresponding to $n$ noise $(\mathbf{x}_i)$ and $n$ data $(\mathbf{y}_j)$ vectors, we follow [Eq.9, Thornton & Cuturi 2022] to initialize $\mathbf{f}, \mathbf{g}$ for a **new** batch $\mathbf{x}'_i, \mathbf{y}’_j$ as $\mathbf{f}^{\text{new-init}}$, computed adaptively as
>
> $$\mathbf{f}^\text{new}_i = 𝜀 \log \tfrac{1}{n} + \min_𝜀(-\langle\mathbf{x}'_i,\mathbf{y}_j\rangle - \mathbf{g}_j), \text{ and setting } \mathbf{g}^{\text{new-init}}=\mathbf{0}_n$$
>
> For ImageNet-32, we observe substantial `~1.7 x` speedups for large $n$ as reported below. We report runtime in seconds, we have checked that using iterations results in the same trends.
>
> ### Imagenet 32, Average Sinkhorn time (in seconds) to solve Alg. 1
> | 𝜀 | $n=$16384 | 65536 | 262144 | 524288 |
> |--:|--:|--:|--:|--:|
> | 0.003 | 5.97 |223.93 |2300.03 |9207.89 |
> | 0.01| 2.02 | 73.64 | 710.4|2893.43 |
> | 0.03|0.65 | 22.01 | 218.63 | 836.82 |
> | 0.1 | 0.18 |5.8|61.25 | 229.85 |
> | 0.3 | 0.09 |2.37 |18.32 |66.73 |
>
> ### Same as above, _**using warmstarting**_
>
> |𝜀|$n=$16384|65536|262144|524288|
> |--:|--:|---:|---:|--:|
> |0.003|3.71|133.54|1271.23 |4916.29|
> |0.01|1.4|48.68|466.47|1791.55|
> |0.03|0.49|16.09|153.78|600.5|
> |0.1|0.14|3.16 |31.75|126.1|
> |0.3|0.06|1.72 |17.6|67.5|
>
> Please note that warmstarting is not an approximation, it is simply a way to leverage previous solutions to solve Alg. 1 faster.
>
> ## 2. Computing matchings in PCA Space
>
> To fit computations for large $n$ in memory, our implementation reinstantiates the cost $\mathbf{C}=-\mathbf{X}^T\mathbf{Y}$ (L.190) block by block at each Sinkhorn iteration (L. 201), as proposed originally in the `GeomLoss` package, later in `OTT-JAX`.This is needed because at $n=10^6$, storing $\mathbf{C}$ is infeasible, even across GPUs (L.204): A `1M x 1M` cost matrix of `float32` would require 4 Tb of memory. As a consequence, the compute cost **per Sinkhorn iteration** is $O(d n^2)$.
>
> When _both_ $n$ and $d$ are huge, we propose to alleviate this by projecting both noise and data onto the top $k$ PCA subspace of _data_ to consider the cost $\mathbf{C}=-\Sigma\mathbf{X}^T(\Sigma\mathbf{Y})$, where $\Sigma$ is $\mathbb{R}^{k\times d}$. This results in per iteration cost of $O(k n^2+ knd)$. Note that FM training (Step. 5-8 in Alg.2) still happens in the **original** space, since PCA is only used to compute pairings.
>
> When applying PCA to **ImageNet-64**,
> * with $n=131,072$ and $𝜀=0.1$, we see up to `10x` speedup over `full` dimension, with no impact on $\mathcal{E}$ nor downstream FM metrics.
> * we can increase $n=1,048,576$ using $𝜀=0.01$ (rightmost cols), which was not feasible with full dimensionality, and achieve even better FID for low NFE.
>
> |PCA dimension $k$ |500|1000|3000|12288 `full`||500|1000|
> |-:|-:|-:|-:|-:|-:|-:|-:|
> |($n, 𝜀$)|(131,072-0.1)|(131,072-0.1)|(131,072-0.1)|(131,072-0.1)| |(1,048,576-0.01)|(1,048,576-0.01)|
> |Sinkhorn time (s)|1.45|1.82| 4.05| 14.1| |445|727|
> |FID@NFE=4|48.4|47.8|47.2|47.2| |46.7|45.7|
> |FID@NFE=8|24.4|24.1|23.8|23.8| |23.9|23.5|
> |FID@NFE=16|15.2|15.1|15.2|15.2| |15.1|15.0|
> |FID@Dopri5|8.44|8.37|8.69|8.64| |8.26|8.53|
> |Ren. Entr. $\mathcal{E}(\mathbf{P}^𝜀)$| 0.247 | 0.239 | 0.232 | 0.236 | | 0.072| 0.072|
>
> *We use the entire dataset to estimate FID statistics for this table, hence the numbers are more accurate than the original draft.*
> ## 3. Precomputing Noise/Data Pairs
>
> Our implementation, to be added to OTT-JAX upon publication, can now precompute pairs of noise / data, as claimed in L.143: We can now decouple Steps 1~4 of Alg. 2 from FM training (Steps 5-8).
> * As $n$ data points are retrieved from a dataloader `DL`, and $n$ Gaussian noises are resampled, the outputs of Steps 1-4 (Sinkhorn + categorical sampling) are accumulated and buffered in a new augmented dataloader `DL~`.
> * To avoid storing noise vectors, we generate each noise vector using $n$ random integer `rng` keys; Rather than store pairs of large vectors in `DL~`, we accumulate the output of Step 4 as pairs of data identifier `id_{i}` seed `rng_{𝜎(i)}` and in `DL~`.
> * When training FM (Step 5-8), we load pairs of indices from `DL~`. For each `rng, id` pair retrieved from `DL~`, the corresponding data vector is retrieved and the noise vector is regenerated using the `rng`.
>
> We use this approach to ablate any hyperparameter of FM training, e.g. learning rate as discussed in the answer to Reviewer **3ETv**, avoiding Sinkhorn recomputations.
>
> _[J. Thornton and M. Cuturi, Rethinking Initialization of the Sinkhorn Algorithm, AISTATS23]_

---

> > ### Comment · Reviewer_zUcn · 2025-08-02
> >
> > I thank the authors for their detailed answers. Most of my questions are resolved, but I would like to emphasize on the following point.
> >
> > The results for $n < 2048$ (the authors showed $32$, $256$, but perhaps including $512$ and $1024$ would be good) should be included in the main text, not the appendix. Without this information, the picture remains incomplete, and it becomes unclear when is min-batch OT helpful. The reasons outlined by the authors as to why mini-batch OT might fail for small $n$, even resulting in worse results, are of great importance.
> >
> > If the authors could include this complete study, I believe this would be an immense contribution to the community.

---

> ### Author Response · Authors · 2025-08-02
> **Many thanks for your acknowledgement! We commit to including very small $n$ results / discussion.**
>
> Many thanks for acknowledging our rebuttal. Many thanks for your encouragement and for your detailed recommendations.
>
> **Adding small $n=256$ results**: We agree that adding these small $n=256$ results (which result in $n=32$ datapoints per device when running on a 8 GPU node) in the paper would be beneficial for the community.
>
> Initially, we did not report these numbers because they were out of range for $\mathcal{R}$ in the $d=256$ column of our piecewise affine synthetic benchmark (see our Table above). We commit to:
> - Report numbers for all dimensions and benchmarks in Fig. 10 / 12, and for $d=256$ do in the same way we report the IFM🔻when it is out of range in the y-axis (in Fig. 10), with an arrow and numerical values, *or*, possibly, use a broken y-axis (we toyed with this idea but gave up at the time due to its complexity, we will try again).
> - Shorten the discussion above on sampling with/without replacement and add it to the main text
> - Run $n=256$ for ImageNet 32 and ImageNet64.
>
> **In the interest of providing additional perspectives on this specific point around ultra-small $n$**: we believe that very small $n$ scales not only yields worse FM metrics, but also brings additional challenges in our implementation which is tailored for larger scales:
> - **Small $n$ incurs a lot of compute overhead**: to take a bit of an extreme argument, while running 8192 times $n=256$ small different Sinkhorn problems is much faster on paper than running a single $n=2,097,152$ problem, there is significant time wasted on GPU with our implementation when reinstantiating these problems, leading to irregular / inefficient GPU utilization, even when considering arguably the most compute intensive setting for 𝜀=0.003
>
> ### Average GPU Utilization, Piecewise Affine Benchmark
> | 𝜀 | $n=$256 | $n=$2,097,152 |
> |----------:|---------:|----------:|
> | 0.003 | 35.82\% |90.66\% |
>
> - Some of our speed-up improvement, such as warmstarting and PCA, won't work as they explicitly leverage a better quality of the dual solution / dimensionality reduction, obtained as $n$ grows.
> - For instance, if we provide the two tables above **[Imagenet 32, Average Sinkhorn time (in seconds) to solve Alg. 1]** and **[Same as above, using warmstarting]** in a single _**speed-up**_ view, and include the smaller $n=2048$ (we initially omitted that column above because we are missing a point), we see that warmstarting is often _detrimental_ for the smallest $n$.
> ### ImageNet-32, speedup obtained when using warmstarting over naive implementation (>1 = better)
> | 𝜀 | $n=$2048 | $n=$16384 | $n=$65536 | $n=$262144 | $n=$524288|
> |----------:|---------:|----------:|----------:|-----------:|-----------:|
> | 0.003 | 0.09 |1.61 |1.68 | 1.81 | 1.87 |
> | 0.01| 1.3|1.44 |1.51 | 1.52 | 1.62 |
> | 0.03| 0.77 |1.32 |1.37 | 1.42 | 1.39 |
> | 0.1 | 0.9|1.36 |1.84 | 1.93 | 1.82 |
> | 0.3 | -|1.4|1.38 | 1.04 | 0.99 |
>
> - This degradation for smaller $n$ is intuitive: because the sample size $n$ is small, the dual potentials estimated by Sinkhorn do not generalize well to unseen data (specially for smallest 𝜀) and fail to provide a useful initialization for the next batch.
>
> We are extremely grateful for your time reading these our draft and our comments. We are thankful for your recommendations. Thanks again!
>
> The authors

---

> > ### Comment · Reviewer_zUcn · 2025-08-02
> >
> > I thank the authors for their commitment to improving the paper. I have increased my score.

---

> ### Author Response · Authors · 2025-08-03
> **Many thanks for your score increase.**
>
> We are very grateful that you read our 2nd response, as well as for your score increase.
>
> Following your comment on small OT batch size $n$ and our commitment to include such experiments in the paper, we are happy to report our preliminary experiments with small $n=256$ on ImageNet-32.
>
> These preliminary experiments confirm that while using small OT batch sizes $n=256$ can offer _some_ FID gains over IFM for the smallest NFEs, it may also perform worse relative to IFM for larger NFE/Adaptive solvers. On the other hand $n=256$ is always _significantly_ worse than large $n$.
>
> The following table shows our experiment with small OT batch size $n=256$ (i.e. $32$ images per device) on ImageNet-32, and compares it with IFM and a larger OT batch size. As seen from the table,
>
> * large OT batch size $n= 524288 $ is substantially better than small $n=256$ for all metrics.
> * For large NFE, we even see that IFM edges above $n=256$ regardless of 𝜀.
>
> | $n$ OT Batch Size | 𝜀 | FID@NFE=4 | FID@NFE=8 | FID@NFE=16 | FID@Dopri5 | Checkpoint (steps) |
> |----|----- |----|-----|----|----|---|
> | IFM           | NA      | 65.8      | 23.9      | _12.1_     | _5.38_     | 180K               |
> | 256           | 0.3     | 42.1      | 20.2      | 13.8       | 8.48       | 180K               |
> | 256           | 0.1     | _39.7_    | _19.4_    | 13.4       | 8.16       | 180K               |
> | 256           | 0.03    | _39.7_    | 19.7      | 13.4       | 8.15       | 180K               |
> | 524288        | 0.03    | **30.2**  | **14.9**  | **9.54**   | **5.18**   | 180K               |
>
> Please note that we make the comparison at the 180K step checkpoint (out of $\approx 430K$ planned) since our runs with $n=256$ batch size haven’t finished yet. However, measures like FID@NFE=4/8 have already plateaued and will not improve at later checkpoints (see also our response to Reviewer **3ETv** showing that ImageNet32 metrics tend to plateau from 150k iterations, and might even slightly increase again after ~250k iterations)
>
> We are seeing very similar trends in ImageNet64 (if not worse), but we need to wait longer to see the runs conclude.
>
> Many thanks again to you (and Reviewer **52g8**) for suggesting this experiment, we agree that it completes our message by also looking at what happens at the "left end of the range", not just at very large $n$.

---

### Public Comment · ~marco_cuturi1 · 2025-10-29
**Unfair decision**

Our paper received five positive (**4,4,4,5,5**), high-quality & confident reviews. With 5 engaged reviewers, the rebuttal was both fruitful and extremely time-consuming not only for authors, but also for reviewers. Meanwhile the AC/SAC both stayed silent, and did not reply to our private messages.

All reviewers unanimously supported the paper, and wrote final recommendations (post-rebuttal) that supported acceptance in very clear terms.

The AC / SAC / PCs rejected the paper arguing two weaknesses listed in the metareview:

▪️ **"the paper is highly technical. it reads more like an engineering/technical report than a scientific paper."**

This criticism is vague, not actionable, and was raised by none of the reviewers. The paper lacks or has too much of what, to be deemed technical vs. scientific? Such a comment would be deemed weak even for a regular review.

▪️ **"The entire methodology is inapplicable to modern conditional problems (e.g., text-to-image generation), where minibatches cannot be constructed in the same way (as for a single condition there is typically just one target)."**

This impossibility claim is baffling: it is confidently laid out, yet weakly supported by a naive reasoning ("cannot be.."). Most problematically, it is wrong.

A direct refutal of this naive claim was in our rebuttal, in which we cited works that used OTFM for conditional generation, notably https://arxiv.org/pdf/2403.18705v3 (JMLR) that was 18 months old at the time of decision. We used the simple method proposed in that paper to provide class conditional generation results in response to Reviewer `zUcn`.

The approach is simple and works for continuous conditions: pair samples of `(noise, random condition in train)` with samples of `(data, ground-truth condition)` using an augmented cost. Reviewer `zUcn` (the only reviewer to mention conditional generation in their review) was satisfied with our answer. Reviewer `zUcn` followed up with unrelated questions & increased their score to **5**.

Additional papers that *apply* and discuss OT couplings methods to *conditional* FM include https://arxiv.org/abs/2404.04240 (**Neurips 24**), https://arxiv.org/abs/2503.10636 (**ICCV 25**).

Note that the AC / SAC / PCs are not expressing a scientific opinion (or doubt) on the feasibility / practicality / scalability of using OT for conditional FM (which we could have discussed), they are blindly stating that the method is "inapplicable". This shows that the AC / SAC / PC chain that validated this meta review is not knowledgeable enough in this area, and they should have exerted caution.

🔺 In summary, in spite of unanimous acceptance ratings (reached after hundreds of hours of work and compute for the rebuttal), the AC/SAC/PCs rejected our paper arguing weaknesses that are poorly phrased or wrong.

Why?

We reached out to the PCs. When confronted to the evidence above, their final argument, after a lengthy exchange, was that unfair decisions happen, and that, while accountable, they do not feel responsible for final decisions. That nuance is lost on me. This is an excuse of last resort.

Remarkably, the final request in the metareview to add connections to rectified flows (a costly **post-processing** approach to distill a high-quality pretrained flow model to make it straighter, which cannot be compared with the **pre-processing** approach presented here to train a FM model from scratch) or Schrödinger bridges resonate as defensive actions by an AC or SAC that would rather silence papers that threaten their research agenda than simply bow to the opinion of reviewers and admit they may not hold all cards.

Ultimately, the PCs sided with the opaque motivations and gate-keeping mindset of the AC/SAC, while ignoring both authors' and reviewers' colossal efforts. This is shameful, and symptomatic of the lack of accountability of the Neurips program committee.

It also shows a wider disregard for authors' efforts that was witnessed at Neurips *this year*, as seen through many disruptive changes to established practices that penalized systematically authors (e.g. last minute change to the pdf rebuttal policy, refusal to allow for links to compensate for this, extensions of rebuttal period, inability for authors to reach out to reviewers if no answer, limitations on number of messages, last minute request to authors to write "Author final remarks" for the AC)

---

### Note · Authors · 2025-08-12

Dear AC, SAC and Reviewers,

While we have no specific remarks to make, we are happy to take this opportunity to thank again all reviewers for their interest in our work. We are grateful for their time and engagement during these 2 months.

In particular, we thank all 5 reviewers for their highly valuable, actionable feedback. Their questions and requests have helped us improve our draft. We are thankful that all our answers and proposals were well received, as evidenced by the fact that 4 reviewers mentioned increasing their score.

Here is a short summary of the changes made to answer their remarks:
- Addition of very small $n=256$ OT batch size results for both synthetic experiments and `Imagenet32` [@resp. **zUcn, 52g8**]
- Addition of class conditional results on `Imagenet32` [@resp. **zUcn**]
- Warm-starting Sinkhorn, with experimental evidence that this works  [@resp. **zUcn**, intended for all reviewers]
- Computation of couplings in data-PCA space for very large $n, d$ regimes with no degradation in FM metrics [@resp. **zUcn**, intended for all reviewers]
- Better explain metaloader idea, i.e. the precomputation of matched noise/data pairs, as briefly mentioned in L.143. Code release to carry out this functionality [@resp. **zUcn**, intended for all reviewers]
- Better explain the `std` rescaling for 𝜀 and the fact that $\mathcal{E}(\mathbf{P}^𝜀)\approx 0$ means $\mathbf{P}^𝜀 \approx$ an optimal permutation matrix [@resp. **52g8**]
- Beyond recalling stats literature (including suggested references), add a simple bound analysis that motivates better that large $n$ is needed for OTFM to be sound [@resp. **7qcL**]
- `AFHQ64` cat → wild modality translation experiment [@resp. **Lgwo**]
- Ablation of learning rates for `ImageNet32` and study of FID throughout iterations [@resp. **3ETv**]

More generally, we have incorporated many more small changes / typo corrections as highlighted by the reviewers.

Respectfully,

the authors

---

### Decision · Program_Chairs · 2025-09-17

**Decision:**

Reject

**Comment:**

The authors consider the Flow Matching (FM) algorithm and study its training with minibatch Optimal Transport (OT) plans, computed via the Sinkhorn algorithm. Unlike the existing literature, which focuses on small minibatches (e.g., 256), the authors explore the use of extremely large batches (up to millions of points) and demonstrate that this approach can benefit FM training. However, utilizing such large batches introduces numerical and computational challenges, which the authors address through several technical contributions, including the automatic rescaling of $\epsilon$ in Sinkhorn, a scale-free renormalized coupling entropy, modifications to the transport cost calculation, and a multi-GPU implementation for scaling Sinkhorn. Experiments are conducted on both synthetic ("W2 benchmark" [1]) and real-world data (CIFAR, ImageNet).

The initial reviews were mixed. Primary concerns included the limited range of OT batch sizes (with a focus on large batches), a lack of conditional generation experiments, potentially misleading plots, and the absence of error bars. During extensive discussions, the authors addressed most of these concerns, leading the reviewers to update their scores positively.

In summary, based on the author-reviewer discussion and the meta-reviewer's own reading of the paper, the **positive aspects** are:

- The paper presents a comprehensive study on the use of large batches in Flow Matching, offering practical and useful recipes.

- It explores minibatch OT in FM at an unprecedented scale, employing significant engineering effort and providing clear guidelines.

The **negative aspects** are:

- The paper is highly technical, and one could argue it reads more like an engineering/technical report than a scientific paper.

- The entire methodology is inapplicable to modern conditional problems (e.g., text-to-image generation), where minibatches cannot be constructed in the same way (as for a single condition there is typically just one target).

While the first bullit point is common nowadays, the second negatively distringuishes the current work from many other papers of the same kind (which propose technical improvelements for generative models). However, in private discussion, reviewers additionally suggested that this paper could hold value not only for the generative modeling community but also for the optimal transport community.

Given the above, the meta-reviewer considers this paper to be borderline. During the AC-SAC discussion, it was decided that the paper should be rejected in its current form. It was also noted that the paper would benefit from addressing additional aspects related to other flow matching and broader research, such as:

- **Synergy with other techniques:** Whether the proposed large optimal transport (OT) batch techniques are orthogonal to other engineering-focused FM research (such as [2]), in the sense that they could be applied together to further improve performance. This includes applying iterative flow matching techniques (e.g., rectified flow, ReFlow [3]) on top of the proposed large-batch OT-FM and evaluating whether this yields additional improvements.

- **Connection to Schrödinger Bridge Methods:** Whether the proposed large-batch OT techniques -- specifically Sinkhorn for entropic OT (also known as the static Schrödinger Bridge) -- can improve the performance (e.g., in terms of FID) of methods based on bridge matching for Schrödinger Bridges. While a complete answer may be beyond the paper's scope, it would be relevant to at least mention the connection between the Schrödinger Bridge and entropic OT, along with the relevant literature. One relevant paper [4] has already been cited in the current study.

The authors are also advised to rework their conclusions to make them more transparent and understandable from a state-of-the-art perspective.

[1] Korotin, A., Li, L., Genevay, A., Solomon, J. M., Filippov, A., & Burnaev, E. (2021). Do neural optimal transport solvers work? a continuous wasserstein-2 benchmark. Advances in neural information processing systems, 34, 14593-14605.

[2] Kim, B., Hsieh, Y. G., Klein, M., Ye, J. C., Kawar, B., & Thornton, J. Simple ReFlow: Improved Techniques for Fast Flow Models. In The Thirteenth International Conference on Learning Representations.

[3] Liu, X., & Gong, C. Flow Straight and Fast: Learning to Generate and Transfer Data with Rectified Flow. In The Eleventh International Conference on Learning Representations.

[4] Tong, A. Y., Malkin, N., Fatras, K., Atanackovic, L., Zhang, Y., Huguet, G., ... & Bengio, Y. (2024, April). Simulation-Free Schrödinger Bridges via Score and Flow Matching. In International Conference on Artificial Intelligence and Statistics (pp. 1279-1287). PMLR.